



# Long-term variation of sea ice and its response to thermodynamic factors in the Northwest Passage of the Canadian Arctic Archipelago

Xinyi Shen[1,2,3], Yu Zhang[1,2,3], Changsheng Chen[4,1,2,3], Song Hu[1,2,3]

1 College of Marine Sciences, Shanghai Ocean University, Shanghai, 201306, China;

5      2 International Center for Marine Studies, Shanghai Ocean University, Shanghai, 201306, China;

3 Center for Polar Research, Shanghai Ocean University, Shanghai, 201306, China;

School for Marine Science and Technology, University of Massachusetts Dartmouth, New Bedford, Massachusetts, 02744, USA

*Correspondence to*: Yu Zhang (yuzhang@shou.edu.cn)

**Abstract.** Sea ice conditions in the Canadian Arctic Archipelago (CAA) play a key role in the navigation of the Northwest Passage (NWP). Based on the observed and simulated sea ice concentration and thickness data, we studied the temporal and spatial characteristics of sea ice from 1979 to 2017 in the NWP of the CAA and evaluated the sea ice conditions along the southern and northern routes of the NWP. Against the background of the rapid retreat of Arctic sea ice, the 39-year observed sea ice concentration of the NWP exhibited a relatively large decreasing trend in summer and fall, while heavy sea ice
conditions were maintained in winter and spring, with a slight increasing trend. Consistent with Arctic sea ice, the sea ice extent in the NWP displayed a decreasing trend of -2.34%/10a, with its minimum occurring in 2012. The sea ice thickness in most subregions of the NWP showed a decreasing trend, with the exception of Lancaster Sound. The decreasing trend of sea ice thickness in the NWP was estimated to -0.16 m/10a. Based on the sea ice concentration and thickness, however, the sea ice conditions were heavier along the northern route than the southern route. This study considered both of these routes, and we
selected and evaluated more specific pathways. The correlation results between the sea ice and atmospheric and oceanic thermodynamic factors in the NWP suggested that the thermodynamic factors had a greater impact on sea ice in the summer and fall, and the variations of sea ice concentration were more closely correlated with the thermodynamic factors than sea ice thickness. The sea surface temperature (SST) had a higher correlation with sea ice concentration than surface air temperature (SAT), while SAT exhibited a higher correlation with sea ice thickness than SST. The residual amount of sea ice concentration
and thickness in the fall, associated with the fall SAT and SST, contributed to the formation of sea ice in the following winter and spring.

**Key words**: Northwest Passage, Sea ice concentration, Sea ice thickness, Surface air temperature, Sea surface temperature



## 1 Introduction

The Northwest Passage (NWP) is a route connecting the Pacific Ocean and the Atlantic Ocean between Asia and Europe that

is 9000 km shorter than the Panama Canal route (Howell and Yackel, 2004). Starting along the northern shore of Alaska in the

United States and northwestern Canada, the NWP passes through the Canadian Arctic Archipelago (CAA) and reaches the

Davis Strait. The opening of the NWP will bring huge economic benefits. There are multiple possible NWP routes through the

CAA (Figure 1). The widest route extends from the Beaufort Sea through the M'Clure Strait into Viscount Melville Sound,

the Barrow Strait and Lancaster Sound, and then into Baffin Bay. In this research, we refer to this route as the northern route.

Of the several possible southern routes, the one known as the PWS route is a modification of the northern route. The PWS

runs from the Amundsen Gulf through the Prince of Wales Strait, and then diverts into Viscount Melville Sound. The three

other southern routes all run from the Amundsen Gulf and through the Coronation and Queen Maud Gulf: the first, referred to

as the MCC route, is through M'Clintock Channel and into Viscount Melville Sound, where it joins the northern route; the

second, known as the PS route, runs through Peel Sound and into the Barrow Strait, where it joins the northern route; and the

third, referred to as the PRI route, passes through Prince Regent Inlet, diverting into Lancaster Sound and then into Baffin Bay.

Sea ice conditions exert a dominant impact on ship navigation in the NWP. As a result of global warming, coverage by Arctic

sea ice is significantly decreasing (Parkinson and Comiso, 2013; Zhang et al., 2016b) and the sea ice thickness is also

diminishing (Kwok et al., 2009; Lindsay and Schweiger, 2015). In particular, the Arctic sea ice extent and area dropped to

historic lows of $3.4 \times 10^6$ km$^2$ and $3.0 \times 10^6$ km$^2$ on 13 September 2012 (Parkinson and Comiso, 2013). The yearly averaged

thickness of Arctic sea ice decreased by 65% in less than four decades, from 3.59 m in 1975 to 1.25 m in 2012 (Lindsay and

Schweiger, 2015).

In recent years, with the increase of world economic and trade demands, the NWP has received constant attention. Some

studies on both the sea ice conditions and the navigation conditions in the CAA have been carried out. From 1979 to 2008, the

CAA sea ice extent exhibited a strong decreasing trend in September, at a rate of -8.7%/10a, and the melt season continued to

lengthen (Howell et al., 2009). Previous studies have noted that a warming climate will bring more severe sea ice conditions

in the NWP (Howell et al., 2006; Melling, 2002). With the longer melt season, the multiyear sea ice is imported from the

northern CAA into the NWP (Howell et al., 2008; Howell et al., 2006), and it may continue for several years, affecting the

navigation of the NWP (Howell et al., 2008). The opening of the NWP in 2007, 2008, and 2010 can be attributed in part to the

lack of sea ice import from the M'Clure Strait (Howell et al., 2013).

The sea ice described in these studies, however, was based on the entire CAA region. Only a few specific studies have focused

on the sea ice conditions in the NWP. In addition, the sea ice conditions in the NWP rarely have been examined based on

subregional divisions. Moreover, because of the incomplete spatial and temporal sea ice thickness data in the NWP, in general,

only sea ice concentration has been taken into account in most previous research, which is applicable only to commercial ships

without strong ice-breaking capacity. For ships with strong ice-breaking capacity, sea ice thickness is a key factor. Furthermore, exploration of the driving mechanisms that influence the sea ice variation in the NWP was insufficient in prior research because atmospheric and oceanic thermodynamic factors exert significant effects on the sea ice conditions.

Therefore, in this study, we utilized the remote sensing data of sea ice concentration and sea ice thickness, as well as the simulated sea ice thickness data from a high-resolution sea-ice coupled model, to examine the sea ice conditions in the NWP

over the period 1979–2017. To compare differences of the subregional sea ice conditions in the NWP in detail, we divided the NWP into 10 subregions: Amundsen Gulf, Coronation and Queen Maud Gulf, M'Clintock Channel, Peel Sound, Prince Regent Inlet, Prince of Wales Strait, M'Clure Strait, Viscount Melville Sound, Barrow Strait, and Lancaster Sound (Figure 1). We also evaluated the sea ice conditions of the northern and southern routes based on the sea ice concentration and thickness, and determined the specific pathways in both the northern and southern routes that were applicable for ship navigation. We also

discussed the impacts of atmospheric (surface air temperature, SAT) and oceanic (sea surface temperature, SST) thermodynamic factors on the sea ice conditions.

## 2 Data and Methods

### 2.1 Observed sea ice data

This study used observational sea ice concentration and thickness data. The sea ice concentration data were the 25-km

resolution monthly Bootstrap sea ice concentration data from the National Snow and Ice Data Center (NSIDC) over the period 1979–2017 (Comiso, 2000) (http://nsidc.org/data/nsidc-0079).

The observed sea ice thickness data were from two sources. The first source included the monthly sea ice thickness data derived from CryoSat-2 (Ricker et al., 2014) (https://www.meereisportal.de/en.html) for the period November 2010 to December 2017, with a horizontal resolution of 25 km. The second source included the weekly Arctic sea-ice thickness data from the CS2SMOS

dataset (Ricker et al., 2017) (https://data.meereisportal.de/data/cs2smos_awi/n/) merged with the CryoSat-2 and Soil Moisture and Ocean Salinity (SMOS) data. The horizontal resolution of these data was also 25 km. The valid data coverage areas from the CryoSat-2 and CS2SMOS in the NWP were different and did not completely cover the NWP. In addition, time was also limited by the lack of thickness data from May to September. Therefore, simulated sea ice thickness data that were reasonably validated with observed data were necessary to fill the temporal and spatial gaps for the study of sea ice thickness in the NWP.

### 2.2 Simulated sea ice thickness data

The simulated sea ice thickness data were from the Arctic Ocean-Finite Volume Community Ocean Model (AO-FVCOM) (Chen et al., 2009; Chen et al., 2016; Gao et al., 2011; Zhang et al., 2016a; Zhang et al., 2016b) for the period 1979–2017, with a horizontal resolution as high as 1 km in the CAA. These high-resolution data could be used to study the spatial distribution, as well as the seasonal and long-term variation characteristics, of sea ice thickness in the NWP. The AO-FVCOM was configured with a nonoverlapped triangular grid that could better reproduce complex shorelines and topography in the NWP. In the vertical resolution, we used a hybrid terrain-conforming coordinate with 45 layers. The driving forces included tidal forcing with eight major constituents ($M_2$, $S_2$, $N_2$, $K_2$, $K_1$, $P_1$, $O_1$, and $Q_1$), surface wind stress, net heat flux at the surface plus shortwave irradiance in the water column, surface air pressure gradients, precipitation (P) minus evaporation (E), and river discharges. We conducted a detailed validation and comparison of the AO-FVCOM sea ice thickness data with the Arctic multisource sea ice thickness data, including ICESat-2 satellite data, field drill-hole observations, airborne electromagnetic observations, and sea ice station data (Zhang et al., 2016b). The results revealed that the sea ice thickness data of the AO-FVCOM well captured the spatial distribution, as well as seasonal and interannual variation characteristics, of Arctic sea ice thickness data. In addition, in a comprehensive comparison with six other sea ice models, the AO-FVCOM achieved better sea ice thickness results in terms of smaller data bias and higher correlation with the multisource sea ice thickness observational data.

### 2.3 Atmospheric and oceanic thermodynamic data

In this study, we used the monthly mean SAT and SST reanalysis data from the European Centre for Medium-Range Weather Forecasts (ECMWF, https://cds.climate.copernicus.eu/) (Hersbach et al., 2019) to explore the impacts of atmospheric and oceanic thermodynamics on sea ice conditions in the NWP. The time period of the SAT and SST data was 1979–2017.

### 3. Results

### 3.1 Spatial and temporal variations of sea ice concentration

### 3.1.1 Spatial distribution of sea ice concentration

From 1979 to 2017, the sea ice concentration over the entire NWP displayed significant spatial distribution differences in different months (Figure 2). The NWP was covered by high sea ice concentrations during spring and winter and lower concentrations in summer and fall. The domain was covered completely by a high sea ice concentration close to 1 (i.e., 100% coverage) from December to April, when both routes were closed by sea ice and would not be navigable by ships lacking ice-

breaking capacity. In May, we observed the sea ice concentration to decrease through three gateways: the Amundsen Gulf, Lancaster Sound, and the M'Clure Strait. Sea ice concentration in September was the lowest, with a distribution pattern showing high sea ice concentrations along the northern route and low concentrations along the southern routes. In addition, the northern route displayed high concentrations in the west and low in the east, while the southern routes exhibited low concentrations in the west and east and high concentrations in the center. Only the M'Clintock Channel, Peel Sound, east Prince of Wales Strait, M'Clure Strait, and Viscount Melville Sound had high sea ice concentrations, ranging from 0.64 to 0.75. The mean sea ice concentration was 0.49 in the Barrow Strait. In most areas of the Amundsen Gulf, Coronation and Queen Maud Gulf, Prince Regent Inlet, and Lancaster Sound, the sea ice concentrations varied from 0.13 to 0.32, and some of the Amundsen Gulf had no sea ice cover. After September, the sea ice started to freeze.

### 3.1.2 Temporal variation of sea ice concentration

The sea ice concentration exhibited a decreasing trend across the entire NWP from 1979 to 2017. Compared with the northern route, larger decreasing trends occurred in the southern routes, especially in the PRI route (Figure 3). The maximum decreasing trend was in the Amundsen Gulf subregion (-4.44%/10a), followed by Lancaster Sound (-4.34%/10a). The Prince of Wales Strait had a minimum decreasing trend, with a mean value of -1.34%/10a.

The sea ice concentration varied in the four seasons. During winter, most subregions in the NWP displayed a slight increasing trend. During spring, the sea ice concentration in most subregions of the NWP had a slight increasing trend, which was similar to winter, with the exceptions of the Amundsen Gulf, M'Clure Strait, and Lancaster Sound gateways. In these subregions, the sea ice concentration exhibited decreasing trends, the maximum of which was -1.17%/10a in the Amundsen Gulf.

In contrast, the sea ice concentration strongly decreased in summer and fall, particularly in the southern routes. In summer, the largest variation was located in the Lancaster Sound (-11.26%/10a) and the smallest rate was in the Prince of Wales Strait (-2.41%/10a). In fall, the variation peak shifted to the Amundsen Gulf (-10.44%/10a), and the smallest rate was still located in the Prince of Wales Strait (-2.98%/10a).

### 3.2 Temporal variation of sea ice extent

To further understand the sea ice conditions in the NWP, we also studied the variation of the sea ice extent. The yearly mean sea ice extent exhibited significant differences among the 10 subregions of the NWP (Figure 4). The mean sea ice extent in the NWP was $4.99 \times 10^5$ km$^2$ (94.41% of the total area) from 1979 to 2017. The sea ice extent displayed a significant decreasing trend during this period, at a rate of -2.34%/10a. The minimum sea ice extent occurred in 2012 ($4.53 \times 10^5$ km$^2$, 85.64% of the total area), which was the same year that the minimum sea ice extent occurred in the entire Arctic Ocean. The seasonal cycle



of sea ice extent for the NWP is also shown in Figure 4, revealing that the NWP was covered by sea ice from November to

April during the study period. The sea ice extent started to decrease in May and reached its minimum ($3.51 \times 10^5$ km$^2$, 70.17%

of the total area) in September.

Similar to sea ice concentration, the mean sea ice extent was also greater along the northern route than along the southern

routes. The mean proportion of sea ice extent coverage was >99.00% in Peel Sound (99.35%), the Prince of Wales Strait

(99.29%), Viscount Melville Sound (99.09%), and the M'Clure Strait (99.03%). The mean coverage rates from 90.00 to 99.00%

were in the M'Clintock Channel (99.00%), Barrow Strait (98.78%), Coronation and Queen Maud Gulf (95.35%), Prince

Regent Inlet (94.78%), and Lancaster Sound (91.86%). The lowest rate was 82.38% in the Amundsen Gulf (Figure 4).

The characteristics of the sea ice extent variation differed significantly among the 10 subregions of the NWP. With the

exception of the Prince of Wales Strait, the other nine subregions of the NWP exhibited significant decreasing trends. The sea

ice extent along the southern routes, particularly the PRI route, displayed larger decreasing trends compared with the other

routes. From 1979 to 2017, the sea ice extent showed significant decreasing trends of -4.80%/10a, -4.47%/10a, -3.06%/10a,

and -2.68%/10a in the Amundsen Gulf, Lancaster Sound, the Coronation and Queen Maud Gulf, and Prince Regent Inlet,

respectively. The larger sea ice extent was maintained from 1979 to 1996 in the M'Clintock Channel, Peel Sound, the M'Clure

Strait, Viscount Melville Sound, and Barrow Strait. After 1997, the sea ice extent exhibited a strongly decreasing trend

(M'Clintock Channel: -1.74%/10a; Peel Sound: -0.84%/10a; M'Clure Strait: -0.40%/10a; Viscount Melville Sound: -

1.10%/10a; Barrow Strait: -0.57%/10a). The years with the smallest sea ice extent were 1998 for the Amundsen Gulf and

Prince of Wales Strait; 2006 for Prince Regent Inlet and Lancaster Sound; 2007 for the M'Clintock Channel; 2010 for the

M'Clure Strait; and 2012 for the Coronation and Queen Maud Gulf, Peel Sound, Viscount Melville Sound, and Barrow Strait.

The seasonal cycle of sea ice extent also manifested different patterns. The sea ice extent began to decrease in May in the

Amundsen Gulf, in June in the M'Clure Strait and Lancaster Sound, in July in the Coronation and Queen Maud Gulf and

Prince of Wales Strait, and in August in the M'Clintock Channel, Peel Sound, Prince Regent Inlet, Viscount Melville Sound,

and Barrow Strait. Except for the minimum sea ice extent occurring in August in Lancaster Sound, the minimum values

occurred in September in the other nine subregions. Larger sea ice extent resumed in October in the M'Clintock Channel, Peel

Sound, Prince of Wales Strait, M'Clure Strait, Viscount Melville Sound, and Barrow Strait, and in November in the Amundsen

Gulf, Coronation and Queen Maud Gulf, Prince Regent Inlet, and Lancaster Sound.

### 3.3 Spatial and temporal variations of sea ice thickness

#### 3.3.1 Comparison of sea ice thickness

Because of the temporal and spatial inconsistency and larger bias between the CryoSat-2 and CS2SMOS data, the monthly

mean simulated sea ice thickness in the NWP from the AO-FVCOM was needed to fill the observation gaps. We performed
the comparison of sea ice thickness between the simulated and observed data by interpolating the simulated results at the
observation locations of the CryoSat-2 and CS2SMOS data for the period November 2010 to December 2017 (Figure 5).
Although the two types of observed sea ice thickness were significantly different, such that their absolute difference from
November 2010 to December 2017 was 0.22 m, they exhibited similar monthly variations, as evidenced by the correlation of
0.91, which was larger in spring and smaller in fall. The simulated sea ice thickness captured the variation pattern well (R =

0.87 and 0.96 with the CryoSat-2 and CS2SMOS data, respectively), matching better with the CryoSat-2 data in spring, and
with the CS2SMOS data in fall. The absolute mean differences between the simulation and the CryoSat-2 and CS2SMOS data
were 0.38 m and 0.20 m, respectively. On the basis of this reasonable comparison result in the NWP and the good validation
results obtained previously in the Arctic with the ICESat-2 satellite, field drill-hole, airborne electromagnetism, and sea ice
station data (Zhang et al., 2016a), the simulated data were capable of being utilized to study the long-term variation of sea ice

thickness in the NWP.

#### 3.3.2 Spatial variation of simulated sea ice thickness

From 1979 to 2017, the sea ice thickness of the NWP exhibited significant spatial distribution differences among different
months and subregions (Figure 6). In the NWP, the sea ice thickness was larger during spring, when it had a mean value of
2.21 m, and smaller in late summer and early fall (August–October), when it had a mean value of 0.39 m. The sea ice thickness

was generally larger along the northern route than along the southern routes.
Along the southern routes, from January to June, there was a large spatial difference in the sea ice thickness, with thinner sea
ice in the Amundsen Gulf, Coronation Gulf, Prince Regent Inlet, and Prince of Wales Strait, and thicker sea ice in the Queen
Maud Gulf, M'Clintock Channel, and Peel Sound. In April, the sea ice thickness reached its maximum (2.15 m) in the southern
routes; the thickest ice occurred in the Coronation Gulf (2.39 m); and the thinnest ice appeared in the Amundsen Gulf (1.83

m). From July to December, the sea ice thickness along the southern routes was larger in the central regions and smaller to the
east and west. The sea ice thickness fell to its minimum in September, when the largest mean value was 0.43 m in the
Coronation Gulf, while the mean values were <0.03 m in the Amundsen Gulf, Prince Regent Inlet, and Prince of Wales Strait.



Along the northern route, the sea ice thickness was generally larger from January to June. The sea ice thickness peaked in May at 2.47 m. From July to December, the sea ice was thicker in the west and thinner in the east. The minimum sea ice thickness occurred in September, when the largest mean value occurred in the M'Clure Strait (0.82 m), and the smallest value was in Lancaster Sound (0.03 m).

### 3.3.3 Temporal variation of simulated sea ice thickness

The time series of the yearly averaged sea ice thickness in the NWP from 1979 to 2017 is shown in Figure 7. The results indicated that the multiyear mean sea ice thickness in the NWP was 1.37 m and the sea ice thickness decreased significantly, at a rate of -0.16 m/10a. The smallest sea ice thickness in the NWP occurred in 2016 (0.93 m). The seasonal cycle of sea ice thickness in the NWP suggested that the thickest sea ice occurred in the spring and the thinnest was in the fall (Figure 7). Specifically, the maximum thickness occurred in April (2.26 m) and the minimum in September (0.26 m).

Figure 8 depicts the spatial patterns of the change rate of sea ice thickness in the NWP over the period 1979–2017. The sea ice thickness exhibited a decreasing trend in most subregions of the NWP. The decreasing trend along the northern route was stronger than the trends along the southern routes, with a maximum trend of -0.30 m/10a occurring in the M'Clure Strait. The increasing trend was mainly distributed in Lancaster Sound (0.07 m/10a). Additionally, there were slight increasing trends in the northern Barrow Strait, northern Coronation Gulf, and southern Prince Regent Inlet.

The change rate varied among the four seasons (Figure 8). The decreasing trend was larger in summer and fall and smaller in winter and spring. In all four seasons, the decreasing trend was generally larger along the northern route than along the southern routes. In winter, the maximum decreasing trend was in the M'Clure Strait (-0.22 m/10a). Slight increasing trends were located on the north side of Lancaster Sound and the Barrow Strait, the northern Coronation Gulf, Queen Maud Gulf, and southern Prince Regent Inlet. In spring, the decreasing trend was weaker than in winter. The largest decreasing trend area was still the M'Clure Strait (-0.20 m/10a), and the increasing trend area expanded in spring, especially in Lancaster Sound, which had an increasing trend of 0.19 m/10a.

The sea ice thickness decreasing trend strengthened noticeably in summer and fall. The M'Clure Strait exhibited the largest decreasing trend in summer (-0.41 m/10a) as well as fall (-0.35 m/10a). The increasing trend weakened in summer, with the exception of Lancaster Sound (0.15 m/10a), and in fall, the sea ice thickness across the entire NWP displayed a decreasing trend.

## 4. Discussion

### 4.1 Evaluation of specific pathways along the northern and southern routes

In general, the northern route had heavier sea ice conditions and a smaller decreasing trend of sea ice concentration and extent, whereas the southern routes had lighter sea ice conditions and a more significant decreasing trend. The northern route, however, had a larger decreasing trend of sea ice thickness than the southern routes. In addition, the northern route is the widest, deepest, and shortest route. Therefore, both the southern and northern routes are important and should be taken into consideration when selecting specific pathways that would be applicable to ship navigation. Figure 9 shows the distribution of mean sea ice concentration, thickness, and probability of light sea ice in the NWP from June to October over the period 1979–2017. We defined the probability of light sea ice as the percentage of sea ice concentration and thickness that was smaller than the climatological mean. The choice of specific pathways was based mainly on smaller sea ice concentration and thickness as well as a larger probability of light sea ice.

On the basis of the mean sea ice concentration and its probability of light sea ice distribution, we selected the specific pathway along the northern route to start from the center of M'Clure Strait, travel to the north side of Viscount Melville Sound and the center of Barrow Strait, and finally continue to the south side of Lancaster Sound. For the southern routes, the specific pathway began at the center of the Amundsen Gulf; then travelled to the Coronation Gulf, the south side of the Queen Maud Gulf, and the center of Prince Regent Inlet; and finally continued to the south side of Lancaster Sound.

On the basis of the sea ice thickness and its probability of light sea ice distribution, the specific pathway along the northern route was kept along the north side, from M'Clure Strait to Lancaster Sound. Along the southern routes, the specific pathway was selected to start on the north side of the Amundsen Gulf, cross the north side of the Coronation Gulf and the south side of the Queen Maud Gulf, continue to the north side of the Prince Regent Inlet, and finally travel to the north side of Lancaster Sound.

### 4.2 Impacts of SAT and SST on sea ice concentration and thickness

#### 4.2.1 Change rates of SAT and SST in the NWP

As important components of sea ice conditions, we studied the change rates of the atmospheric (SAT) and oceanic (SST) thermodynamic factors (Figure 10). We further investigated the impacts of SAT and SST on sea ice concentration and thickness and discussed the correlations between these thermodynamic factors and the sea ice conditions (Figures 11 and 12). Note that the interaction among sea ice, ocean, and atmosphere is a complex process. In this study, we considered only the impact of thermodynamic factors on sea ice. Further examination of the interaction between sea ice and thermodynamic factors remains

necessary. Over the period 1979–2017, the change rates of both SAT and SST exhibited increasing trends across the entire NWP (Figure 10). Consistent with the change rate of sea ice concentration (Figure 3), the increasing rates of SAT and SST were larger along the southern routes than the northern route. The maximum increasing trends of SAT (8.87%/10a) and SST

(50.84%/10a) were located in the Amundsen Gulf. There were also some differences between the variations of SAT and SST. Compared with SST, the SAT in Peel Sound, Prince Regent Inlet, M'Clure Strait, and Barrow Strait had larger increasing trends.

**4.2.2 Impacts of SAT and SST on sea ice concentration**

We investigated the spatial differences of the correlations between sea ice concentration and thermodynamic factors (SAT,

SST) in different seasons, as shown in Figure 11. The seasonal correlation coefficient results suggested that SAT and SST affected sea ice concentration more significantly in summer and fall than in winter and spring. Meanwhile, SST had closer negative correlations with the variation of sea ice concentration than SAT.

In winter, the sea ice concentration over the entire NWP did not seem to be influenced by either SAT or SST, as indicated by the very small respective mean correlation coefficients of -0.04 and -0.05 (Figures 11a, 11e), because of the nearly unchanged

sea ice concentration over the period 1979–2017 (Figure 13a). For the correlation between sea ice concentration and SAT, its distribution exhibited small negative coefficients in some subregions of the NWP, with the maximum mean coefficient of -0.40 occurring in Lancaster Sound (Figure 11a). The positive coefficients were mainly located in the western section of the northern route. Although the SAT in winter showed an increasing trend from 1979 to 2017, the low temperatures did not affect the sea ice melting (Figure 13a). Because the NWP is fully covered by sea ice in winter and the SSTs are mostly induced from

sea ice temperature (Wang et al., 2018), the NWP SSTs in winter did not change in any practical sense. For the correlation between sea ice concentration and SST, most subregions of the NWP also displayed no significant correlation, with values ranging from -0.25 to 0.21 (Figure 11e).

In spring, although the sea ice concentration and SST showed slight interannual variations over the period 1979–2017 (Figure 13b), the low sea ice concentration correlations with SAT and SST were maintained in most subregions of the NWP, with the

exceptions of the Amundsen Gulf, M'Clure Strait, and Lancaster Sound gateways (Figures 11b, 11f). The correlation of sea ice concentration exhibited a larger mean correlation coefficient with SST (-0.52) than with SAT (-0.35) in the three gateways. The maximum sea ice concentration correlations with SAT and SST were both located in the Amundsen Gulf. The correlations with the significant spatial differences were related to the early melting in the three gateways in May and June (Figure 2).

The correlation of the entire NWP between sea ice concentration and SAT, SST increased rapidly in summer and fall. In

summer, because of the significant increasing trend of SAT and SST and decreasing trend of sea ice concentration (Figure

13c), the correlation coefficient of sea ice concentration with SAT ranged from -0.55 to -0.81 and -0.47 to -0.86 with SST. The correlation was higher along the southern routes than the northern route (Figures 11c, 11g). In fall, the high correlation of sea ice concentration continued in the subregions, ranging from -0.51 to -0.74 for SAT and from -0.48 to -0.89 for SST, with the higher correlations still located along the southern routes (Figures 11d, 11h). Although the increasing trend of SST was reduced

in fall (Figure 13d), the SST exhibited a higher mean correlation coefficient (-0.81) than SAT (-0.67) across the entire NWP.

### 4.2.3 Impact of SAT and SST on sea ice thickness

For the thermodynamic factors, the correlation coefficients of sea ice thickness were smaller than those of the sea ice concentration. In general, SAT had a higher correlation with sea ice thickness than SST in all four seasons. The correlation distribution also illustrated the significant seasonal differences, with higher correlations in summer and fall and lower

correlations in winter and spring (Figure 12).

In winter, SAT exhibited a higher negative correlation (-0.26) with sea ice thickness than SST, because the variations of SAT and sea ice thickness were more significant (Figure 13a). The sea ice thickness and SAT displayed negative correlations in most subregions of the NWP, with the maximum coefficient appearing in the Amundsen Gulf (Figure 12a). In contrast, the correlation coefficients between sea ice thickness and SST showed weak positive values in most of the NWP, with the

exception of M'Clure Strait (Figure 12e).

In spring, SAT still affected the sea ice thickness more than SST. Most subregions in the NWP showed negative correlations between sea ice thickness and SAT, whereas positive correlations were mainly distributed in Barrow Strait, Lancaster Sound, and some areas of Queen Maud Gulf and Prince Regent Inlet (Figure 12b). These findings were consistent with the increasing trend of sea ice thickness during spring (Figure 8). The correlation between sea ice thickness and SST was not significant in

most subregions of the NWP. In addition, compared with winter, the positive coefficients extended to the M'Clintock Channel, Peel Sound, and Viscount Melville Sound (Figure 12f).

In summer, the sea ice thickness correlations with SAT and SST increased sharply. The spatial distributions of the correlations with SAT and SST were similar, with mean values of -0.43 and -0.45, respectively. With the exception of Lancaster Sound, the subregions in the NWP exhibited negative correlations, with coefficients between -0.24 and -0.54 (Figure 12c) for SAT

and -0.21 and -0.61 for SST (Figure 12g). The correlations were higher along the southern routes than the northern route, with a maximum in the Coronation and Queen Maud Gulf (-0.54) for SAT and the Amundsen Gulf (-0.61) for SST.

Compared with summer, the correlation coefficients in fall increased, with a mean value of -0.61 for SAT and -0.46 for SST. The sea ice thickness and SAT, SST displayed negative correlations across the entire NWP (Figures 12d, 12h). The sea ice thickness correlation with SAT increased in most areas of the NWP, with coefficients ranging from -0.49 to -0.75. The

correlations along the southern routes were higher than the northern route, especially along the PRI route. The maximum coefficient was located in the Amundsen Gulf. The sea ice thickness correlation with SST was lower than with SAT, ranging from -0.33 to -0.58 in the subregions. The correlations along the southern routes were higher than the northern route, especially along the PRI route, with the maximum in the Amundsen Gulf.

From this discussion, it appeared as though the sea ice thicknesses in winter and spring were not influenced significantly by

SAT and SST. The relationships between seasonal mean sea ice concentration and thickness in fall and the sea ice thickness in the following winter over the period 1979–2017 are presented in Figure 14. These results suggested that the residuals of sea ice concentration and thickness from the fall had more significant impacts on the sea ice thickness in the following winter. The correlations between sea ice concentration and sea ice thickness in fall and the sea ice thickness in the following winter were 0.82 and 0.93, respectively, indicating that larger or smaller sea ice concentration and thickness in fall generally corresponded

to larger or smaller sea ice thickness in the following winter, respectively. For example, the maximum sea ice concentration and thickness in the NWP during the fall occurred in 1979, which then resulted in the maximum sea ice thickness in the following winter. Conversely, in 1998, the sea ice concentration in the fall experienced a significant drop, from 0.73 in 1997 to 0.52, as did the sea ice thickness, from 1.20 m in 1997 to 0.69 m. The following winter, the sea ice thickness also decreased sharply, from 1.80 m in 1997 to 1.44 m. Additionally, the impact of sea ice concentration and thickness in fall could extend to

the sea ice thickness in the following spring, as indicated by the corresponding correlation coefficients of 0.76 and 0.80. Because the residual sea ice concentration and thickness in fall was associated with the concurrent SAT and SST, higher thermodynamic values could cause more sea ice melting. This suggested that SAT and SST in fall affected the sea ice thickness in the following winter and spring by controlling the residual sea ice concentration and thickness in the fall.

## 5. Conclusions

Sea ice conditions exerted a significant impact on ship navigation in the NWP of the CAA. Because of the insufficient observed data, previous studies of the NWP mainly focused on the sea ice concentration, whereas few studies have examined the variation of sea ice thickness, which is the other key characteristic of sea ice conditions. Furthermore, few studies have explored the impacts of the atmospheric and oceanic thermodynamic factors on the sea ice conditions in terms of subregions. In this study, we used the combined observed data of sea ice concentration and sea ice thickness, as well as the simulated sea ice

thickness from a high-resolution sea-ice coupled model over the period 1979–2017, to investigate sea ice conditions, including sea ice concentration and thickness, in the subregions of the NWP. We evaluated the specific pathways of both the northern and southern routes for ship navigation, taking into consideration both the sea ice conditions and light sea ice probability in



the NWP from June to October. In addition, the different impacts of SAT and SST on sea ice conditions in the subregions were examined and discussed by analyzing the seasonal correlations and comparing the interannual variations.

The seasonal spatial distributions of sea ice concentration varied significantly. In winter and spring, the entire NWP was fully covered by a sea ice concentration that was close to 1. During summer and fall, the sea ice concentration decreased differently in different subregions. In general, the sea ice concentration was greater along the northern route than the southern routes. The 39-year observed sea ice concentration of the NWP exhibited relatively large decreasing trends in summer and fall, whereas the heavy sea ice conditions were maintained in winter and spring, with a slight increasing trend, which differed from the

decreasing sea ice concentration trend in all four seasons across the entire Arctic. The sea ice extent in the NWP displayed a significant decreasing trend (-2.34%/10a), with the minimum occurring in 2012 ($4.53 \times 10^5$ km$^2$, 85.64% of the total area), a pattern that was consistent with the sea ice extent over the entire Arctic.

From 1979–2017, the interannual variation of sea ice thickness decreased at a rate of -0.16 m/10a, reaching its minimum in 2016 (0.93 m). The multiyear mean seasonal sea ice thickness in the NWP increased from October to April, with its maximum

of 2.26 m occurring in April and its minimum of 0.26 m occurring in September. In general, the sea ice thickness along the northern route was greater than the thicknesses along the southern routes, while the decreasing trend of the northern route was stronger. In the most areas of the NWP, with the exception of Lancaster Sound, the sea ice thickness exhibited a decreasing trend, which was larger in summer and fall and smaller in winter and spring.

Based on the sea ice concentration and thickness distribution from June–October and the probability of light sea ice, which

was defined as the percentage of lighter sea ice conditions compared with the climatological means, we evaluated additional specific pathways along the southern routes and the northern route. According to sea ice concentration, the more specific pathway for the southern routes was the center of the Amundsen Gulf-Coronation Gulf–south side of the Queen Maud Gulf–center of Prince Regent Inlet–south side of Lancaster Sound. For the northern route, the more specific pathway was the center of M'Clure Strait–north side of Viscount Melville Sound–center of Barrow Strait–south side of Lancaster Sound. Considering

sea ice thickness, the more specific pathway for the southern routes was the north side of the Amundsen Gulf–north side of the Coronation Gulf–south side of the Queen Maud Gulf–north side of Prince Regent Inlet–north side of Lancaster Sound. For the northern route, the more specific pathway was to travel along the north side from M'Clure Strait to Lancaster Sound.

As for the impacts of atmospheric and oceanic thermodynamic factors on sea ice conditions, the correlations between sea ice and thermodynamic factors were higher in summer and fall, and the sea ice concentration generally exhibited a higher

correlation than the sea ice thickness. The SST had a higher correlation with sea ice concentration than SAT, whereas the SAT had a higher correlation with sea ice thickness than SST. The sea ice thickness in the winter and spring was dominant by the residual of sea ice concentration and thickness, which was affected by SAT and SST in the previous fall.

**Acknowledgments**

This work was supported by National Natural Science Foundation of China 41706210, National Key Research and
Development Program of China 2019YFA0607000, 2016YFC1400903 for Yu Zhang, 2018YFC1406801 for Song Hu, and
the US NSF grant PLR-1603000 for Changsheng Chen.

**Data availability**

The sea ice concentration is available from the National Snow and Ice Data Center (NSIDC) (http://nsidc.org/data/nsidc-0079).
The sea ice thickness data was obtained from CryoSat-2 (https://www.meereisportal.de/en.html) and the CS2SMOS dataset
(https://data.meereisportal.de/data/cs2smos_awi/n/). The surface air temperature (SAT) and sea surface temperature (SST) was
obtained from the European Centre for Medium-Range Weather Forecasts (ECMWF, https://cds.climate.copernicus.eu/).
Other data that support the findings of this study are available as described in the Methods and otherwise from the
corresponding author upon request.

**Author Contributions**

Xinyi Shen and Yu Zhang collected observational data. Yu Zhang and Changsheng Chen provided the simulated data. Xinyi
Shen performed data processing. Xinyi Shen, Yu Zhang and Song Hu conducted data analysis. Xinyi Shen summarized key
finds and wrote the manuscript. Yu Zhang, Song Hu and Changsheng Chen revised the manuscript.

**Competing interests**

The authors declare that they have no known competing financial interests or personal relationships that could have appeared
to influence the work reported in this paper.

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

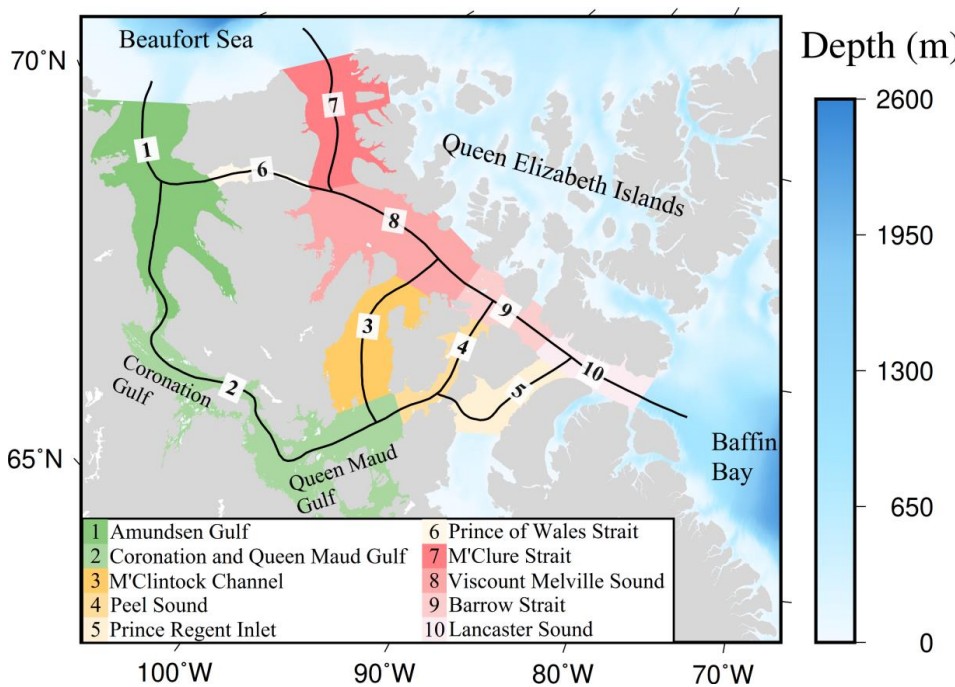


**Figure 1. Bathymetry of the Canadian Arctic Archipelago and 10 subregions of Northwest Passage routes.**





**Figure 2. Monthly mean sea ice concentration distribution in the NWP from 1979–2017.**



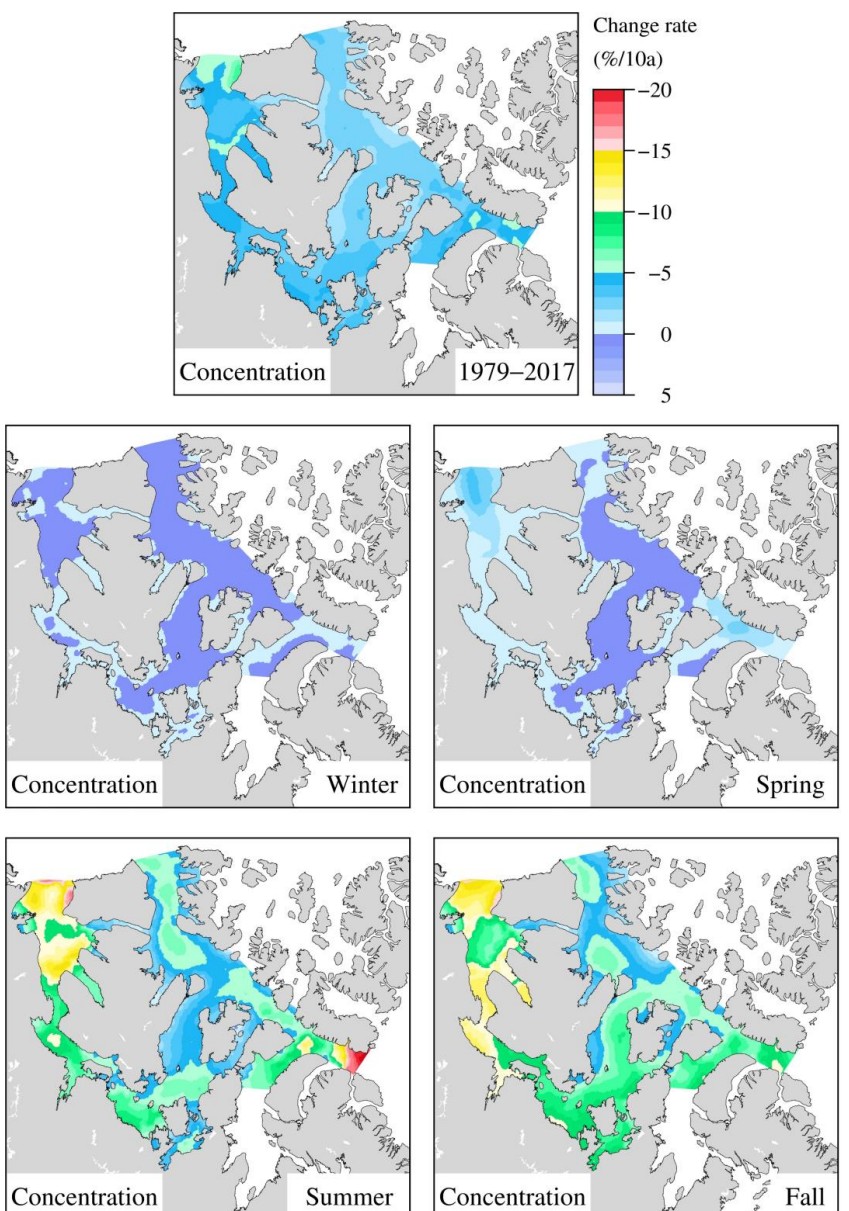

**Figure 3. Multi-year and seasonal mean change rate distribution of sea ice concentration from 1979–2017.**





**Figure 4. Yearly mean sea ice extent (blue dots) for the NWP and 10 subregions of the NWP from 1979 to 2017. Red lines indicate the linear regression trends. The subfigures inserted in the lower left of each panel illustrate the seasonal variability of the sea ice extent (blue plot) over the period 1979–2017.**





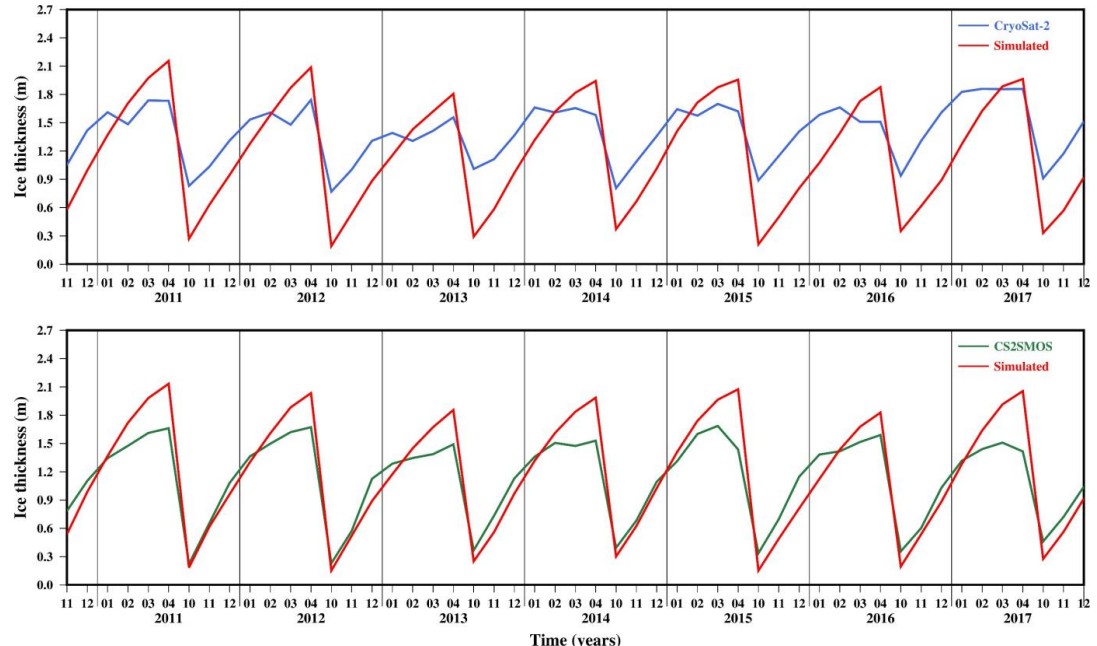


**Figure 5. Monthly sea ice thickness from the CryoSat-2 (blue curve), CS2SMOS (green curve), and AO-FVCOM (red curves) in the NWP from November 2010 to December 2017.**





Figure 6. Monthly mean sea ice thickness distribution from 1979–2017.



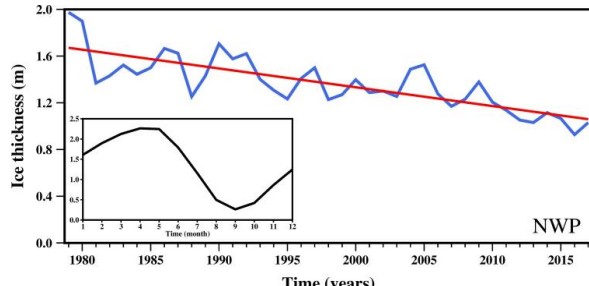


**Figure 7. Yearly sea ice thickness (blue plot) in the NWP from 1979 to 2017. The red line indicates the linear regression trend. The subfigure inserted in the lower left shows the seasonal variability of the sea ice thickness (black curve) over the period 1979–2017.**





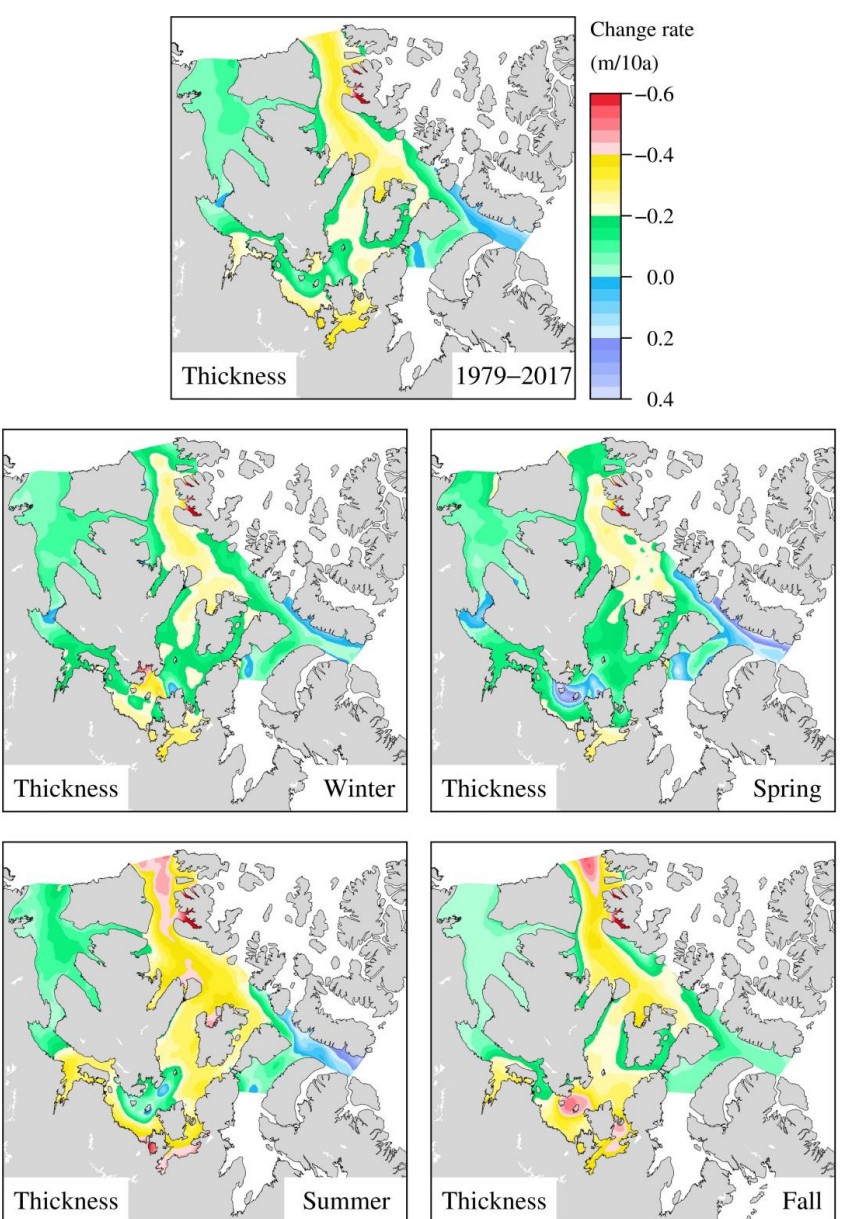

**Figure 8. Multiyear and seasonal mean change rate distribution of sea ice thickness from 1979 to 2017.**



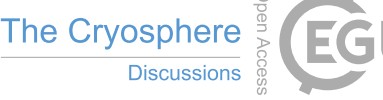

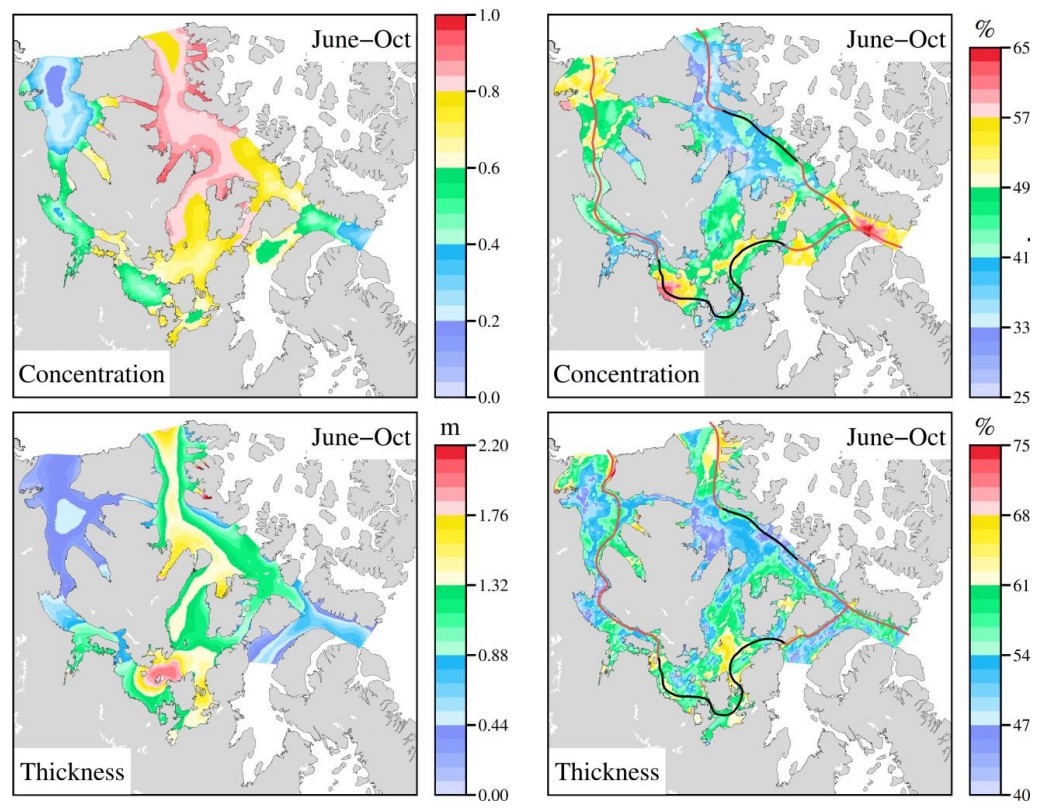


**Figure 9. Distribution of mean sea ice concentration and thickness (left panel) and probabilities of light sea ice (right panel) in the NWP from June to October. The black and red curves respectively indicate the same and different specific navigation pathways between the evaluation of sea ice concentration and thickness.**

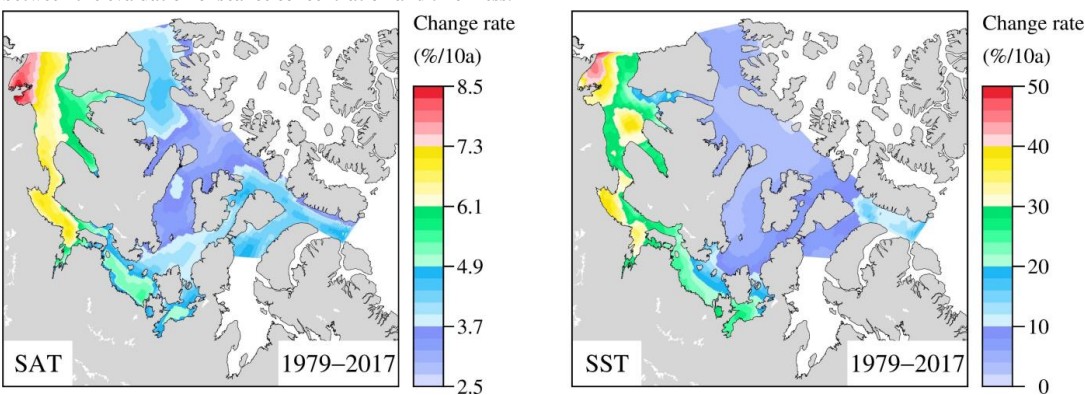

**Figure 10. Distribution of change rates of SAT and SST in the NWP from 1979 to 2017.**





**Figure 11. Distribution of seasonal correlation coefficients between sea ice concentration and SAT and SST in the NWP.**





**Figure 12. Distribution of seasonal correlation coefficients between sea ice thickness and SAT, SST in the NWP.**





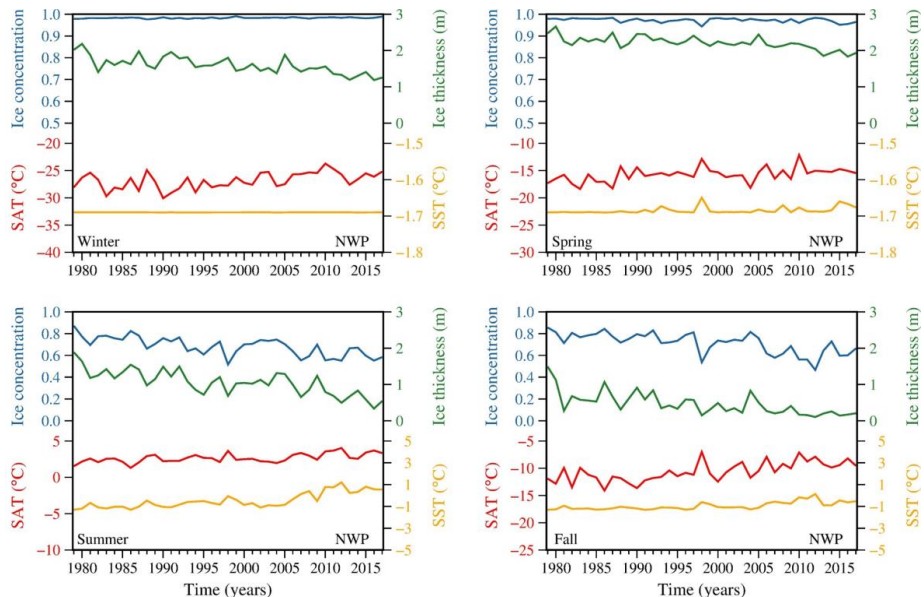


**Figure 13. Variations of seasonal mean sea ice concentration (blue), sea ice thickness (green), SAT (red), and SST (orange) in the NWP from 1979 to 2017.**

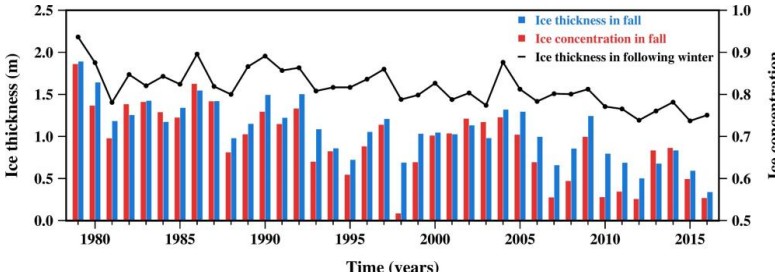

**Figure 14. Relationships of seasonal mean sea ice thickness (blue) and concentration (red) in the fall and sea ice thickness (black) in**

**the following winter from 1979 to 2016.**