# Peer review of "Long-term variation of sea ice and its response to thermodynamic factors in the Northwest Passage of the Canadian Arctic Archipelago"

_The Cryosphere, 2020_

## Referee Comment (RC1) · Anonymous Referee #1 · 3 Sep 2020

Long-term variation of sea ice and its response to thermodynamic factors in the Northwest Passage of the Canadian Arctic Archipelago

Shen and others investigated the spatiotemporal characteristics of sea ice extent and thickness in the Northwest Passage regions of the Canadian Arctic Archipelago from 1979-2017 and also looked at the forcing links from SAT and SST. The Northwest Passage does represent an important region worthy of scientific investigation but there are significant problems with this manuscript that are as follows:

The first problem is the quality of the data used in this study which has major implications for the results they present and the conclusions they draw. The qual-

[Figure]

ity of AO-FVCOM sea ice thickness estimates within the CAA has not been assessed/validated and therefore it is unknown how much uncertainty there is and how much the results can be trusted. I looked at the Zhang et al. (2016b) JGR paper and I noticed all the in situ measurements were outside the CAA so in fact, there was no sea ice thickness validation done in the CAA. There are in fact in situ measurements of ice thickness as well as airborne EM induction data which could be used for validation (e.g. Haas and Howell, 2015; Howell et al., 2016). Furthermore, there have been major recent assessments of model performance in the CAA which have not been cited (e.g. Howell et al., 2016; Kushner et al., 2018; Laliberté et al., 2018). These uncited studies the authors have missed relate the problem with state-of-the-art climate models having difficultly resolving sea ice thickness within the CAA (Howell et al., 2016; Laliberté et al., 2018). In short, the authors have not provided enough evidence to state that the AO-FVCOM is any better than other models whereby large over estimation in the trends was found to be problematic in the CAA. Another data quality problem not discussed by the authors is from altimetry sea ice thickness estimates within the CAA. Sea ice thickness retrievals from satellite altimetry are highly uncertain within the majority of the CAA (certainly the NWP) because there no leads (see Landy et al., 2017).

The second problem is the lack of new information on sea ice conditions within NWP. For example, the authors are incorrect to state that "only a few studies have focused on the sea ice conditions in the NWP", "sea ice conditions in the NWP rarely have been examined based on subregional divisions" and "only sea ice concentration has been taken into account in most previous research." Almost every study the authors cite and the numerous they do not cite (because they missed a lot) all do this (e.g. Howell et al., 2008; Tivy et al., 2011; Derksen et al., 2012; Haas and Howell, 2015; Mudryk et al., 2018) and as a result, the justification for this study very weak and then when I got to the results I found that there was not really any new information that is not already known. For example, they boldly state that "Furthermore, exploration of the driving mechanisms that influence the sea ice variation in the NWP was insufficient in prior research because atmospheric and oceanic thermodynamic factors exert significant

effects on the sea ice conditions." I do not think the author's have immersed themselves in the literature sufficiently to make this bold statement and their results are certainly less rigorous than previous studies (see Tivy et al., 2011 for links to SAT). Moreover, there is not one reference to previous studies in Section 3-4 of this manuscript and there is a vast body of work on CAA trends/variability which should have at least been compared to. Another is example of a lack of understanding is evident when discussing why the correlation between SAT and sea ice thickness is low. The author's completely ignored (or missed) that the reason is because snow thickness has been found to be the primary driver of ice thickness within the CAA not SAT (see Brown and Cote, 1992 and Howell et al., 2016).

References:

Brown, R. and Cote, P (1992), Interannual variability of landfast ice thickness in the Canadian high arctic, 1950–89, Arctic, 45, 273–284.

Derksen, C., S. Smith, M. Sharp, L. Brown, S.E.L. Howell, Y. Gauthier, D. Mueller, L. Copland, A. Tivy, R. Brown, C. Fletcher, C. Burn, A. Lewkowicz, C. Duguay, M. Bernier, P. Kushner and A. Walker (2012), Variability and change in the Canadian cryosphere. Climatic Change, 115(1), 59-88, doi: 10.1007/s10584-012-0470-0.

Haas, C., and S. E. L. Howell (2015), Ice thickness in the Northwest Passage, Geophysical Research Letters., 42, 7673–7680, doi:10.1002/2015GL065704.

Howell, S.E.L., A. Tivy, J.J. Yackel, and S. McCourt (2008), Multi-year sea ice conditions in the Western Canadian Arctic Archipelago section of the Northwest Passage: 1968-2006. Atmosphere-Ocean, 46(2), 229–242 doi:10.3137/ao.460203.

Howell, S. E. L., F. Laliberté, R. Kwok, C. Derksen and J. King (2016), Landfast ice thickness in the Canadian Arctic Archipelago from Observations and Models, The Cryosphere, 10, doi:10.5194/tc-2016-71

Kushner, P. J., L.R. Mudryk, W. Merryfield, J.T. Ambadan, A. Berg, A. Bichet, R. Brown,
C. Derksen, S.J. Déry, A. Dirkson, G. Flato, C.G. Fletcher, J.C. Fyfe, N. Gillett, C. Haas, S.E.L. Howell, F. Laliberté, K. McCusker, M. Sigmond, R. Sospreda-Alfonso, N.F. Tandon, C. Thackeray, B. Tremblay and F.W. Zwiers (2018), Assessment of Snow, Sea Ice, and Related Climate Processes in Canada's Earth-System Model and Climate Prediction System, The Cryosphere, 12, 1137-1156, https://doi.org/10.5194/tc-12-1137-2018.

Laliberté, F., S.E.L. Howell, J-F. Lemieux, F. Dupont and J. Lei (2018),What historical landfast ice observations tell us about projected ice conditions in Arctic Archipelagoes and marginal seas under anthropogenic forcing, The Cryosphere, 12, 3577-3588, https://doi.org/10.5194/tc-12-3577-2018.

Landy, J., J.K. Ehn, D.G. Babb, N. Thériaulta and D G. Barber (2017), Sea ice thickness in the Eastern Canadian Arctic: Hudson Bay Complex & Baffin Bay, Remote Sensing of Environment, 200,281-294 http://dx.doi.org/10.1016/j.rse.2017.08.019

Mudryk, L., C. Derksen, S.E.L. Howell, F. Laliberté, C. Thackeray, R. Sospedra-Alfonso, V. Vionnet, P. Kushner and R. Brown (2018), Canadian Snow and Sea Ice Trends and Projections, The Cryosphere, 12, 1157-1176, https://doi.org/10.5194/tc-12-1157-2018.

Tivy, A., S.E.L. Howell, B. Alt, S. McCourt, G. Crocker, T. Carrieres and J.J. Yackel (2011), Trends and variability in summer sea ice cover in the Canadian Arctic based on the Canadian Ice Service Digital Archive, 1960 to 2008 and 1968-2008. Journal of Geophysical Research-Oceans,116, C03007, doi:10.1029/2009JC005855

---

## Referee Comment (RC2) · Anonymous Referee #2 · 22 Sep 2020

Review of "Long-term variation of sea ice and its response to thermodynamic factors in the Northwest Passage of the Canadian Arctic"

By: Shen et al.,

The Cryosphere – TC-2020-215

This paper examines changes in sea ice concentration and thickness in the southern portion of the Canadian Arctic and attempts to ascribe these changes to thermodynamic forcing from either either surface air temperatures or sea surface temperatures. A mix of observed sea ice concentration and modelled ice thickness is used, while air temperature and sea surface temperature were retrieved from an ERA- reanalysis

product. All of this work is done in the context of declining sea ice and the opening of the Northwest Passage for marine shipping. While I think there is some interesting work done on sea ice concentration and thickness trends in the CAA, the manuscript suffers from oversimplification of key details on both shipping and declining sea ice because I think it is trying to cover too much and looks at too coarse of a time period. The paper would benefit from a detailed discussion of shipping through the NWP and how declining sea ice has led to increased shipping through the NWP over the last decade. In particular this would highlight the fact that shipping (non ice-strengthened, or ice-strengthened, non-ice-breakers) can only occur under very specific ice-free conditions during a short window in late-summer. The presentation of shipping pathways based on the probability of sea ice being thinner or less concentrated than the historical mean isn't practical. Even if the ice cover is thinner than it was historically, it is still very likely too thick for shipping to occur along the NWP. In terms of sea ice, again there are some interesting results, but the section correlating sea ice conditions to thermodynamic factors is very confusing and doesn't reveal a clear outcome. The authors present a lot of information and have included some nice analysis, but I think the objectives of the paper need to be clarified before the paper is revised.

Considering the major revisions I encourage the authors to make I would suggest they focus on the major comments first. I have attached minor comments as well, but considering the paper will likely be heavily revised I suggest they leave these minor comments until later.

Major Comments: 1. Shipping along the NWP is given as a motivator for this work, but there is very little actual discussion of shipping along the NWP. I would suggest that the authors provide a detailed introduction to shipping along the NWP that discusses its benefits (shorter route), its limitations (sea ice), the recent increase in ships along the NWP, its seasonality (which is key), and the projected potential for shipping along the NWP in a warming Arctic. In particular I would recommend the authors look at the following list of works and really strengthen the motivation for this work. • Piz-

zolato et al., (2014), Changing sea ice conditions and marine transportation activity in Canadian Arctic waters from 1990 and 2012, Climatic Change, 123, 161-173, doi: 10/1007/s10584-013-1038-3. • Pizzolato et al., (2016), The influence of declining sea ice on shipping activity in the Canadian Arctic, Geophysical Research Letters, 43, doi: 10.1002/2016GL071489. • Melia, Haines and Hawkins (2016), Sea ice decline and 21st century trans-Arctic shipping routes, GRL, 43, doi: 10.1002/2016GL069315 • Ng, Andrewsm Babb, Lin, Becker (2018), Implications of climate change for shipping: Opening the Arctic Seas, WIRES. • Dawson et a., (2018), Temporal and Spatial patterns of ship traffic in the Canadian Arctic from 1990 to 2015, Arctic, 71(1), doi: 10.14430/arctic4696

Based on this revised discussion I think the discussion of shipping pathways needs to be heavily revised or removed. Basing a pathway on the probability of sea ice being lighter than the historical mean isn't realistic, because an ice cover that is thinner than the mean may still be too thick for a majority of vessels to travel through. Instead, I would suggest using polar codes or literature on arctic shipping to define thresholds and then examine when ice conditions that meet those thresholds exist. This would be much more practical, but is likely moving away from your thermodynamic analysis and more into the realm of shipping focused papers.

2. For all of the correlation and trend analysis, I would suggest only presenting significant ($p < 0.05$) values. This would highlight real changes and remove some questionable results like small trends towards increasing sea ice concentration during winter and spring.

3. With respect to the Cryosat2 ice thickness data. Since it is only used to quickly assess the accuracy of the modelled ice thickness then I would suggest moving this discussion and Figure 5 to your methods sections. I would also suggest only using the CS2SMOS product as it is much more accurate over areas of thin ice (which you note with its accuracy during fall) and as opposed to presenting the comparison in a time series, present it as a scatterplot of monthly means. Further to this comment and this

section a. L 81-82: What areas of the NWP did the CS2SMOS product not cover? b. L 82 - 83: It's worth noting here why there is no ice thickness data from Cryosat-2 during the melt season. c. L83-84: Beyond saying the modeled ice thickness was "reasonably validated" please provide an exact measure of correlation or bias here. Also consider rewording as you don't use the modelled ice thickness to "fill the temporal and spatial gaps" but you instead use it throughout your full analysis.

4. The correlation section is very difficult to understand and doesn't provide a clear result. I also think it needs to be reinforced that this is a thermodynamic analysis and as Howell has shown in several papers, dynamics, particularly the transport of multiyear ice within the CAA and along the NWP is an important process.

Minor Comments:

L 12: This comment applies throughout the paper, but when referring to your study region it is the Canadian Arctic Archipelago and the NWP runs through it. I would suggest revising this sentence to read "... we studied the temporal and spatial characteristics of sea ice from 1979 to 2017 in the CAA and evaluated the sea ice conditions along the southern and northern routes of the NWP".

L 14-15: The term "heavy" ice conditions isn't really clearly defined, so I would suggest revising to "the region remained ice covered throughout winter and spring during this period". Additionally based on my suggestion to present only significant trends, I think the text about there being a slight increasing trend can be removed.

L 17: I don't see evidence from Figure 3 of increasing SIC in Lancaster Sound. Please check this statement

L 18: replace "Based on the sea ice concentration and thickness, however the sea ice conditions ..." with "Generally, sea ice conditions were heavier along the northern route than the southern route, with a longer ice season and thicker ice".

L 20-24: I have other comments about the correlation analysis that will likely cause this

text to be revised. But when your revising this please be specific in your statements. An example is "...Thermodynamic factors had a greater impact on sea ice in the summer and fall, than during winter and spring".

L 24-26: I don't think "residual" is the correct word for this. You're talking about the remaining ice that persisted through summer and already exists at the start of fall freeze-up. Also this remaining ice is not only influenced by fall SST and SAT, but also summer SST and SAT. I think this statement needs to be revised.

L 30-34: Back to one of my major comments, but this introductory text can be strengthened. The NWP connects Europe and Asia > it is shorter than the Panama Canal Route, but historically it has been ice covered and unsafe for marine vessels. However, as ice declines the NWP is becoming increasingly accessible... I would then reference the works of Pizzolato and Dawson about increase shipping activity along the NWP.

L 33: "The opening of the NWP will bring huge economic benefits", please provide a reference for this and specify who will benefit? Also what about the additional risk for communities and the environment around the NWP?

L 34 – 35: Remove "the" from in front of M'Clure Strait and Barrow Strait.

L 36: Note right away that there are 3 southern routes that all rely on Lancaster Sound and Amundsen Gulf but pass through different channels in the central part of the CAA. Essentially, this description of the routes can be tightened up.

L 48-55: This text on sea ice in the CAA is good, but to the point of warming increasing ice severity along the NWP, I think it should be noted that MYI enters the northern CAA from the Arctic Ocean and migrates through to the southern CAA during summers as the ice cover opens up. Additionally, Haas and Howell (2015) observed modal thicknesses of 1.8 and 2.0 m along the NWP with deformed MYI having a mode of 3.0 m. This would be good to include so you can refer back to it later when presenting your modeled ice thicknesses. Also it would be worth noting the previous minima in 1998

and 2007 described by Howell et al., (2010) • Haas and Howell (2015), Ice thickness in the Northwest Passage, GRL, 42, doi: 10.1002/2015GL065704. • Howell, Tivy, Agnew, Markus, Derksen (2010), Extreme low sea ice years in the CAA: 1998 versus 2007, JGR, 115, doi: 10.1029/2010JC006155.

When discussing sea ice within the CAA it needs to be clear that the ice is mobile and there is a mix of first year and multiyear sea ice in the CAA. Specifically the Drain trap mechanism for MYI in the central part of the CAA described by Howell et al., 2008 should be described. As well as the fact that ice is imported and exported from the three gateway regions (Amundsen, M'Clure and Lancaster), particulary during spring and summer.

L 59-60: This connects back to a previous comment, but when expanding the discussion of shipping through the NWP note the difference between the open water shipping season and potential for ice-breakers. The difference in vessels is critical for shipping along the NWP.

L 63: revise "we utilize a combination of remotely sensed sea ice concentration data and modelled ice thickness data to examine the sea ice conditions . . .."

L 68-69: There's a comment below about this selection of a ship path, but I think this needs to be explained in more detail and presented as the "optimal" or "route through the lightest ice conditions".

Methods:

L 74-76: Please elaborate on the description of this dataset. Howe is it collected and what are its limitations. In particular passive microwave data is known to underestimate sea ice concentration during the melt period. This error should be consistent through the time series so it wont dramatically affect your results, but it should be discussed.

Regarding the interpretation of sea ice concentration data, you commonly refer to extent, but I believe you are calculating sea ice area. This is good, but the figure labels

and text need to be revised. Also please add a sentence about this in the methods.

Regarding sea ice concentration data in Prince of Wales Strait – this is a fairly narrow channel and with 25 km resolution I'm wondering how many actual sea ice pixels are contained in this channel and how reliable that data is. Also I haven't seen this channel discussed as part of the NWP before, typically there is one of the two southern routes or the northern route through M'Clure.

L 88: resolution is up to 1 km , but what is the range?

L 95: It is just ICESat, not ICESat-2.

L 94 – 100: In this text please note that all of this work was presented by Zhang et al., (2016). Adjust the start of this text from "We conducted. . ." to "Zhang et al., (2016) validated ice thickness from AO-FVCOM with a multisource dataset. . .".

L 102-104: Which ERA reanalysis did you use? -Interim or -5? Please specify.

Results: L 108 and throughout: revise the text "significant spatial distribution differences" to " significant differences in the spatial distribution" or "significant spatial differences". As it's written it is tricky to read.

L116 – 120: I'm not sure these means are really worth presenting given the significant negative trends you are about to present in the next section. I think the 1979-2017 mean shows the general pattern of ice loss, but I wouldn't get too focused on the actual values.

L 120: "After September, the sea ice started to freeze", this is pretty informal, I'd suggest adding some more detail here.

L 124-125: Again check that these trends are significant. The Prince of Wales Strait is not significant, so this text needs to be revised. Also just a note that the figures show sea ice extent (area) and the trends are presented. Perhaps provide both the trend in extent and then provide the % for further context.

[Figure]

L 126-127: back to one of the major comments, but I am really suspicious of this increasing trend during winter. There may be some variability, but that is likely due to the error of passive microwave sea ice concentration retrievals and not real.

L 143 – 147: With respect to Figure 4, the key takeaway is the negative trends in each sub-region. I don't think discussing the mean annual sea ice concentration in each region over the 38 year record is that useful, especially given the substantial changes taking place.

L 153: Replace "The larger sea ice extent" with "A near complete ice cover..."

L 154: The change that occurred around 1997 is related to the 1998 minimum discussed by Howell et al., 2010. Please add the reference of 1998 to the introduction and then you can refer back to it here. Howell, Tivy, Agnew, Markus, Derksen, 2020, Extreme low sea ice years in the Canadian Arctic Archipelago: 1998 versus 2007, GRL.

L 163: "Larger sea ice extent resumed in October..." this is a part of fall freeze-up. Beyond listing the regions, I think it would be more useful to state that freeze-up begins earlier in the central and northern part of the CAA (M'Clintock, Peel, Prince of Wales, M'Clure, Viscount, Barrow) in October and then expands to the southern and peripheral part of the CAA during November.

Section 3.3.1: See one of the major comments above about removing this section and moving the brief validation of the model to the methods.

L 178: Remove the 2 from ICESat-2.

L 182: revise " significant spatial distribution differences"

L 183 – 184: As opposed to saying "the sea ice thickness was larger in spring and small in late summer and early fall", simply say " the sea ice was thicker in spring and thinner in later-summer and early fall".

L 186 – 187: Thicker ice is located in the Queen Maud Gulf, M'Clintock Channel, and Peel Sound because it operates as a drain trap for Multiyear ice within the CAA as described by Howell et al., 2008. Please add discussion of this to your introduction so you can refer back to it here. Howell, Tivy, Yackel, McCourt (2008), Multi-year sea-ice conditions in the western Canadian Arctic Archipelago Region of the northwest passage: 1968-2006, Arctic.

L 203 and Figure 8: Again please only display and discuss the significant trends.

L 205 – 206: With respect to the increasing trend in the Labrador Strait, first is this significant? If so what is the mechanism for this?

L 223: "deepest" I think this is in reference back to Figure 1, but the bathymetry of the CAA isn't actually shown in that figure. Please revise or add a reference to this point.

L 226-228: The process of selecting pathways based on "light sea ice conditions" needs to be revised and considerably improved. Thinner ice doesn't necessarily make a route passable of the ice is still relatively thick and therefore hazardous for all but a few ships. This goes back to the first major comment, a better introduction of shipping along the NWP and the focus on the open water shipping season is needed before this discussion is suitable. Also, if basing the route on the change relative to the historical mean, its important to note the historical mean represents a concentrated, thick ice cover that didn't break up during summer.

Section 4.2.2: I find this section to be very confusing and difficult to understand. I would encourage you to clarify this discussion. In particular I would suggest you only focus on significant correlations. Additionally, it's important to remember that this only reflects thermodynamic forcing and not dynamics, which Howell has shown to be key for the CAA in several of his works.

L263-264: "the low temperatures did not affect the sea ice melting", I'm not sure what is meant by this statement.

L 264-265: How are the SST's observed during winter? Are they simply set to be at the freezing point when under sea ice? This should be discussed in the methods and will likely cause this text to be revised.

L 311: "Suggest" instead of "suggested"

L 311: I'm not sure "residuals" is the right word for what you're describing here. This is the state of the ice cover after the September minimum and at the start of fall-freeze-up.

L 321-323: Connecting the state of the ice cover in fall to the state of the ice cover at the end of winter is interesting, but the last sentence about fall SAT and SST affecting sea ice thickness the following winter is a little misleading, because the state of the ice cover in fall is predominantly dictated by the summer conditions and not just fall.

Conclusions:

Based on the comments above it seems that the conclusions may be revised considerably, but here are some minor comments to handle now.

L 325: remove the "ed" from "exerted".

L 329: Based on a previous comment, you don't really use observed sea ice thickness data. It's all from the model.

L 336 and in other places within the paper: Please be consistent and present sea ice concentration as a percentage (%) as opposed to just "1".

L 339: Another instance the "increasing trend" during winter. Please check that this is significant.

L 343: revise the first sentence to read " from 1979-2017, sea ice thickness in the NWP decreased…"

L 344: Revise " The multiyear mean seasonal sea ice thickness" to read " The monthly mean sea ice thickness…"

L 347-348: Remove "the" from "In the most. . ." and "Lancaster Sound, the sea ice. . .".

L 348: In the conclusions provide a value for these trends.

L 361: Revise the word "dominant", the end of winter ice cover is influenced by the fall ice cover, but I don't think it is the dominant factor.

Figures:

Figure 1: Note that you don't actually show bathymetry in the CAA so I would suggest removing bathymetry from this figure.

Figure 3: Is the top panel the trends in the annual mean sea ice concentration? I would suggest removing that and focusing on the seasonal means.

Figure 4: For the inset of annual cycles can you provide some bounds of the standard deviation or

Figure 5: This figure can very likely be removed based on comments above.

Figure 6: I'd suggest removing "distribution" from the caption as an ice thickness distribution is something other than this figure.

Figure 7: Note in the caption that this is the "annual mean thickness".

Figure 8: Again, only show the significant trends and consider removing the annual mean.

Figure 10: Are these the annual mean changes in SAT and SST? Also, instead of the % please consider revising to show the magnitude (°C) of the trends and only showing the significant trends.

Figure 11 and 12: It is slightly counter intuitive to flip the colorbar so that blue is positive correlations and red is negative. Also, once again consider showing only the significant correlations.

---

## Author Comment (AC1) · 24 Sep 2020

Responses to Reviewer #1

General comment The first problem is the quality of the data used in this study which has major implications for the results they present and the conclusions they draw. 1. The quality of AO-FVCOM sea ice thickness estimates within the CAA has not been assessed/validated and therefore it is unknown how much uncertainty there is and how much the results can be trusted. I looked at the Zhang et al. (2016b) JGR paper and I noticed all the in situ measurements were outside the CAA so in fact, there was no sea ice thickness validation done in the CAA. There are in fact in situ measurements of

ice thickness as well as airborne EM induction data which could be used for validation (e.g. Haas and Howell, 2015; Howell et al., 2016). Furthermore, there have been major recent assessments of model performance in the CAA which have not been cited (e.g. Howell et al., 2016; Kushner et al., 2018; Laliberté et al., 2018). These uncited studies the authors have missed relate the problem with state-of-the-art climate models having difficultly resolving sea ice thickness within the CAA (Howell et al., 2016; Laliberté et al., 2018). In short, the authors have not provided enough evidence to state that the AO- FVCOM is any better than other models whereby large over estimation in the trends was found to be problematic in the CAA.

Answer: - We appreciate the reviewer's helpful comments. We have revised the manuscript to add the new validation using the observed sea ice thickness of Canadian Ice Services (CIS) the reviewer recommends from Howell et al., 2016. There are two sites in our study region which is Cambridge Bay and Resolute. We did the detailed comparison between the simulated results and these two observed sites. The results are shown below. - From 1979 to 2017, AO-FVCOM captured the seasonal variation feature of sea ice thickness in the Cambridge Bay and Resolute (Figure 1). Howell et al., 2016 suggested that some other models overestimated the sea ice thickness in these two sites. For example, the root mean square error (RMSE) between PIOMAS and observed sea ice thickness was 0.29 cm at Cambridge Bay and 0.68 cm at Resolute. The value of our simulated sea ice thickness was also larger than the observed data. However, compared with other models, the RMSE between our simulation and observations was reduced to 0.18 m at Cambridge Bay and 0.52 m at Resolute (Figure 2, Table 1). The seasonal variation of sea ice thickness was also captured well by the simulation. In addition, Howell et al., 2016 compared the trend of maximum sea ice thickness and found that the simulated result showed larger trend. We also did the same analysis and the results was reasonable (Table 1). In the Resolute, both observation and simulation showed very close decreasing trend of maximum sea ice thickness with the value of -0.07 m/10a and -0.06 m/10a. In the Cambridge Bay, maximum sea ice thickness exhibited observed and simulated decreasing trend of -0.07 m/10a and

[Figure]

-0.12 m/10a, respectively. This difference was also smaller than the models Howell et al., 2016 mentioned.

2. Another data quality problem not discussed by the authors is from altimetry sea ice thickness estimates within the CAA. Sea ice thickness retrievals from satellite altimetry are highly uncertain within the majority of the CAA (certainly the NWP) because there no leads (see Landy et al., 2017).

Answer: - We agreed with the reviewer's comment and we will revise the manuscript to add the discussion and related reference about the quality problem and uncertainty of satellite altimetry sea ice thickness. This would help the readers to further understand the satellite data.

The second problem is the lack of new information on sea ice conditions within NWP. 1. For example, the authors are incorrect to state that "only a few studies have focused on the sea ice conditions in the NWP", "sea ice conditions in the NWP rarely have been examined based on subregional divisions" and "only sea ice concentration has been taken into account in most previous research." Almost every study the authors cite and the numerous they do not cite (because they missed a lot) all do this (e.g. Howell et al., 2008; Tivy et al., 2011; Derksen et al., 2012; Haas and Howell, 2015; Mudryk et al., 2018) and as a result, the justification for this study very weak and then when I got to the results I found that there was not really any new information that is not already known. For example, they boldly state that "Furthermore, exploration of the driving mechanisms that influence the sea ice variation in the NWP was insufficient in prior research because atmospheric and oceanic thermodynamic factors exert significant effects on the sea ice conditions." I do not think the author's have immersed themselves in the literature sufficiently to make this bold statement and their results are certainly less rigorous than previous studies (see Tivy et al., 2011 for links to SAT). Moreover, there is not one reference to previous studies in Section 3-4 of this manuscript and there is a vast body of work on CAA trends/variability which should have at least been compared to. Another is example of a lack of understanding is evident when discussing
why the correlation between SAT and sea ice thickness is low. The author's completely ignored (or missed) that the reason is because snow thickness has been found to be the primary driver of ice thickness within the CAA not SAT (see Brown and Cote, 1992 and Howell et al., 2016).

Answer: - We apologize some statements cause the misunderstanding for the reviewer. For the statements reviewer mentioned, the original purpose is to suggest that only a few studies selected the NWP as the only research domain. The references the reviewer recommended cover the whole CAA and the NWP is only one of their research domains. That is the reason why we did not cite these references. We appreciate the reviewer's helpful comment and we will add the references, remove the inappropriate words and revise the manuscript to avoid the misunderstanding. - Due to the limit of observed sea ice thickness in the NWP, the understanding of temporal and spatial variation of sea ice thickness needs to be enhanced. Since this study is only focused on the NWP, it provides a more detailed study of long-term variation of sea ice condition by dividing the NWP into ten major subregions. The results of sea ice thickness in these subregions including the completed temporal and spatial variation, the distribution of seasonal change rate and the relation with sea surface temperature and surface air temperature were introduced and discussed separately. Additionally, the specific shipping routes along the northern and southern routes were evaluated and selected with the consideration of both sea ice concentration and thickness data. These findings could give us further insight into the understanding of sea ice condition in the NWP. - In addition, we followed the reviewer's suggestion that we will add the references and compared the previous studies of CAA trends/variability with ours. - For the correlation between sea ice concentration and thickness with SAT and SST (section 4), we will add more discussion to explain the reason why the correlation between sea ice thickness and SAT is low.

We are working on the revised manuscript and we will attach a draft revision and highlight the revised places. After getting the comments from other reviewers, we
will make further revision. Hopefully, these answers and revisions could meet the reviewer's requirement.

Please also note the supplement to this comment:
https://tc.copernicus.org/preprints/tc-2020-215/tc-2020-215-AC1-supplement.pdf
* * *
[Figure]

[Figure]

**Fig. 1.** Figure 1. Seasonal variability of the sea ice thickness from CIS (red curve) and AO-FVCOM (blue curves) over the period 1979–2017.

[Figure]

**Fig. 2.** Figure 2. Comparison of AO-FVCOM sea ice thickness with sea ice thickness observations over the period 1979–2017.

| | Cambridge Bay | | Resolute | |
|---|---|---|---|---|
| | Observations | Simulation | Observations | Simulation |
| Mean sea ice thickness (m) | 1.32 | 1.38 | 1.41 | 1.73 |
| Trend of sea ice thickness (m/10a) | -0.07(p<0.05) | -0.12(p<0.01) | 0.03 | 0.00 |
| Trend of maximum sea ice thickness (m/10a) | -0.07(p<0.05) | -0.12(p<0.01) | -0.07(p<0.05) | -0.06 |
| Mean absolute differences (m) | 0.10 | | 0.33 | |
| Correlation coefficient | 0.96(p<0.01) | | 0.77(p<0.01) | |
| RMSE (m) | 0.18 | | 0.52 | |

**Fig. 3.** Table 1. Mean sea ice thickness, trend and maximum sea ice thickness trend of observations and AO- FVCOM in the Cambridge Bay and Resolute, and mean absolute differences, correlation coefficient, RMSE

**Supplement:**

[Figure]

**Figure 1. Seasonal variability of the sea ice thickness from CIS (red curve) and AO-FVCOM (blue curves) over the period 1979–2017.**

[Figure]

**Figure 2. Comparison of AO-FVCOM sea ice thickness with sea ice thickness observations over the period 1979–2017.**

**Table 1. Mean sea ice thickness, trend and maximum sea ice thickness trend of observations and AO-FVCOM in the Cambridge Bay and Resolute, and mean absolute differences, correlation coefficient, RMSE between observations and AO-FVCOM.**

|  | Cambridge Bay | | Resolute | |
| --- | --- | --- | --- | --- |
|  | Observations | Simulation | Observations | Simulation |
| Mean sea ice thickness (m) | 1.32 | 1.38 | 1.41 | 1.73 |
| Trend of sea ice thickness (m/10a) | -0.07($p<0.05$) | -0.12($p<0.01$) | 0.03 | 0.00 |
| Trend of maximum sea ice thickness (m/10a) | -0.07($p<0.05$) | -0.12($p<0.01$) | -0.07($p<0.05$) | -0.06 |
| Mean absolute differences (m) | 0.10 | | 0.33 | |
| Correlation coefficient | 0.96($p<0.01$) | | 0.77($p<0.01$) | |
| RMSE (m) | 0.18 | | 0.52 | |

---

## Referee Comment (RC3) · Anonymous Referee #3 · 28 Sep 2020

Long-term variation of sea ice and its response to thermodynamic factors in the Northwest Passage of the Canadian Arctic Archipelago author by Shen et al. The manuscript (MS) is interesting and fits with the scope of the journal but unfortunately, the data and the interpretation are not well presented. As the authors have highlighted in the MS title "response to thermodynamic factors", but fail to justify the factors. The authors have discussed only the relation with SST and SAT. To understand thermodynamics, we should know the mixed layer depth (MLD), then only we could know the ocean heat transport. In my opinion, the article cannot be published in that form needs a lot of substantial improvements and modifications: I, therefore, suggest the article cannot be accepted in the present form.

[Figure]

The abstract is very simple and doesn't show the novelty of the pertaining long-term sea ice and its response to thermodynamic factors. After reading the abstract I could see authors have just given the decadal observation of SIC and their correlation with SST and SAT. This section is drafted very poorly with the unfocused aim and finding highlights. Although the approached techniques are good but not justified by the authors in their explanations. Suggested to be more focused and rewrite the abstract. This section is lacking with clear aim and objective of the work as well as the concluding remarks/novelty of the work. Need to be more specific about the computational and processing techniques.

The sea ice thickness data for the Canadian Arctic Archipelago were utilized from for the model output of the AO- FVCOM. I could not see any data validation and any specific reasons for choosing this model. If any such study may be given. Data and methods are not complete need to be elaborated properly. I could not find any analysis details. Sea ice extent data details and analysis are missing how SIE was calculated? How the authors have divided the NWP into 10 subregions? What was the criteria or reference have been considered to divide the CAA? Materials and methods are poorly written and incomplete.

Authors have represented their results in just quantitative way in terms of spatial and temporal changes of SIC, SIE and SIT although sea-ice parameters have been published earlier by several authors sector-wise of whole Arctic regions. The MS is lacking with process and mechanism. Authors have attempted to explain the variations with only SST and SAT, this study needs to be extended by considering ocean heat transport and budget. The sea ice declining processes and their forcings are must be highlighted. In this present form, the paper is not recommended for publication.

---

## Author Comment (AC2) · 10 Oct 2020

Major comments

1. Shipping along the NWP is given as a motivator for this work, but there is very little actual discussion of shipping along the NWP. I would suggest that the authors provide a detailed introduction to shipping along the NWP that discusses its benefits (shorter route), its limitations (sea ice), the recent increase in ships along the NWP, its seasonality (which is key), and the projected potential for shipping along the NWP in a warming Arctic. In particular I would recommend the authors look at the following list of works and really strengthen the motivation for this work. Pizzolato et al., (2014), Changing

sea ice conditions and marine transportation activity in Canadian Arctic waters from 1990 and 2012, Climatic Change, 123, 161-173, doi: 10/1007/s10584-013-1038-3. Pizzolato et al., (2016), The influence of declining sea ice on shipping activity in the Canadian Arctic, Geophysical Research Letters, 43, doi: 10.1002/2016GL071489. â ËŸ A′ c Melia, Haines and Hawkins (2016), Sea ice decline and 21st century trans-Arctic shipping routes, GRL, 43, doi: 10.1002/2016GL069315 Ng, Andrewsm Babb, Lin, Becker (2018), Implications of climate change for shipping: Opening the Arctic Seas, WIRES. â ËŸ A′ c Dawson et a., (2018), Temporal and Spatial patterns of ship traffic in the Canadian Arctic from 1990 to 2015, Arctic, 71(1), doi:10.14430/arctic4696 Based on this revised discussion I think the discussion of shipping pathways needs to be heavily revised or removed. Basing a pathway on the probability of sea ice being lighter than the historical mean isn't realistic, because an ice cover that is thinner than the mean may still be too thick for a majority of vessels to travel through. Instead, I would suggest using polar codes or literature on arctic shipping to define thresholds and then examine when ice conditions that meet those thresholds exist. This would be much more practical, but is likely moving away from your thermodynamic analysis and more into the realm of shipping focused papers.

Answer: We appreciate the reviewer's helpful comments. We will revise the manuscript to add a detailed introduction to shipping along the NWP following the listed works and add the references the reviewer recommended. For the discussion of shipping pathways, we apologize that some statements cause the misunderstanding for the reviewer. We do not aim to choose the shipping pathway which is suitable for the present navigation. As the reviewer mentioned, due to the heavy sea ice condition, the NWP is hard to be used in the most time of the year. Even if using polar codes or literature on arctic shipping to define thresholds, it is still difficult to find a specific pathway for shipping. The specific pathway we selected is mainly about a statistical analysis to examine the locations in the NWP which have larger probability of light sea ice condition based on the historical data. Then we connected these locations together and got the specific pathway which has larger potential opportunity for navigation in the

future. We appreciate the reviewer's helpful comments and we will revise the statement and make it clear.

2. For all of the correlation and trend analysis, I would suggest only presenting significant ($p < 0.05$) values. This would highlight real changes and remove some questionable results like small trends towards increasing sea ice concentration during winter and spring.

Answer: We have revised the pictures (Figure 1-5) and will revise the analysis.

3. With respect to the Cryosat2 ice thickness data. Since it is only used to quickly assess the accuracy of the modelled ice thickness then I would suggest moving this discussion and Figure 5 to your methods sections. I would also suggest only using the CS2SMOS product as it is much more accurate over areas of thin ice (which you note with its accuracy during fall) and as opposed to presenting the comparison in a time series, present it as a scatterplot of monthly means. Further to this comment and this section a. L 81-82: What areas of the NWP did the CS2SMOS product not cover? b. L82 - 83: It's worth noting here why there is no ice thickness data from Cryosat-2 during the melt season. c. L83-84: Beyond saying the modeled ice thickness was "reasonably validated" please provide an exact measure of correlation or bias here. Also consider rewording as you don't use the modelled ice thickness to "fill the temporal and spatial gaps" but you instead use it throughout your full analysis.

Answer: We appreciate the reviewer's comment and we will revise the manuscript. We have removed the comparison between AO-FVCOM and CryoSat2 sea ice thickness and added the comparison between AO-FVCOM and CS2SMOS as a scatterplot of monthly means (Figure 6). In addition, we have added the new comparison with observed sea ice thickness of Canadian Ice Services (CIS) and provided the detailed comparison of mean absolute differences, correlation coefficient and RMSE (Table 1). Through adding the new comparison, we will show the seasonal variation and trend of sea ice thickness in different station. CS2SMOS could not cover the Coronation Gulf,

part of Queen Maud Gulf, Prince of Wales Strait, Peel Sound and part of Barrow Strait and Lancaster Sound. The number of valid data is different in different time. We will add this introduction into the manuscript. We have revised the incorrect statement "fill the temporal and spatial gaps".

4. The correlation section is very difficult to understand and doesn't provide a clear result. I also think it needs to be reinforced that this is a thermodynamic analysis and as Howell has shown in several papers, dynamics, particularly the transport of multiyear ice within the CAA and along the NWP is an important process.

Answer: We will revise the manuscript to add more discussion with previous studies.

Minor comments

1. L 12: This comment applies throughout the paper, but when referring to your study region it is the Canadian Arctic Archipelago and the NWP runs through it. I would suggest revising this sentence to read "... we studied the temporal and spatial characteristics of sea ice from 1979 to 2017 in the CAA and evaluated the sea ice conditions along the southern and northern routes of the NWP".

Answer: Actually, our study region is the area where the NWP runs through the Canadian Arctic Archipelago. We will revise the related description.

2. L 14-15: The term "heavy" ice conditions isn't really clearly defined, so I would suggest revising to "the region remained ice covered throughout winter and spring during this period". Additionally based on my suggestion to present only significant trends, I think the text about there being a slight increasing trend can be removed.

Answer: We will revise the manuscript.

3. L 17: I don't see evidence from Figure 3 of increasing SIC in Lancaster Sound. Please check this statement

Answer: In L17, it is said "The sea ice thickness in most subregions of the NWP showed

a decreasing trend, with the exception of Lancaster Sound". It is about sea ice thickness, not SIC. Please check Figure 2 in this response.

4. L 18: replace "Based on the sea ice concentration and thickness, however the sea ice conditions ..." with "Generally, sea ice conditions were heavier along the northern route than the southern route, with a longer ice season and thicker ice".

Answer: We revised the manuscript.

5. L 20-24: I have other comments about the correlation analysis that will likely cause this text to be revised. But when your revising this please be specific in your statements. An example is "...Thermodynamic factors had a greater impact on sea ice in the summer and fall, than during winter and spring".

Answer: We revised the manuscript.

6. L 24-26: I don't think "residual" is the correct word for this. You're talking about the remaining ice that persisted through summer and already exists at the start of fall freeze-up. Also this remaining ice is not only influenced by fall SST and SAT, but also summer SST and SAT. I think this statement needs to be revised.

Answer: We revised the statement as: The remaining amount of sea ice concentration and thickness in the fall, associated with the SAT and SST in summer and fall, contributed to the formation of sea ice in the following winter and spring.

7. L 30-34: Back to one of my major comments, but this introductory text can be strengthened. The NWP connects Europe and Asia > it is shorter than the Panama Canal Route, but historically it has been ice covered and unsafe for marine vessels. However, as ice declines the NWP is becoming increasingly accessible... I would then reference the works of Pizzolato and Dawson about increase shipping activity along the NWP.

Answer: We will revise the manuscript.

8. L 33: "The opening of the NWP will bring huge economic benefits", please provide a reference for this and specify who will benefit? Also what about the additional risk for communities and the environment around the NWP?

Answer: We will revise the manuscript.

9. L 34 – 35: Remove "the" from in front of M'Clure Strait and Barrow Strait.

Answer: Revised.

10. L 36: Note right away that there are 3 southern routes that all rely on Lancaster Sound and Amundsen Gulf but pass through different channels in the central part of the CAA. Essentially, this description of the routes can be tightened up.

Answer: We revised the manuscript.

11. L 48-55: This text on sea ice in the CAA is good, but to the point of warming increasing ice severity along the NWP, I think it should be noted that MYI enters the northern CAA from the Arctic Ocean and migrates through to the southern CAA during summers as the ice cover opens up. Additionally, Haas and Howell (2015) observed modal thicknesses of 1.8 and 2.0 m along the NWP with deformed MYI having a mode of 3.0 m. This would be good to include so you can refer back to it later when presenting your modeled ice thicknesses. Also it would be worth noting the previous minima in 1998 and 2007 described by Howell et al., (2010) Haas and Howell (2015), Ice thickness in the Northwest Passage, GRL, 42, doi: 10.1002/2015GL065704. Howell, Tivy, Agnew, Markus, Derksen (2010), Extreme low sea ice years in the CAA: 1998 versus 2007, JGR, 115, doi: 10.1029/2010JC006155. When discussing sea ice within the CAA it needs to be clear that the ice is mobile and there is a mix of first year and multiyear sea ice in the CAA. Specifically the Drain trap mechanism for MYI in the central part of the CAA described by Howell et al., 2008 should be described. As well as the fact that ice is imported and exported from the three gateway regions (Amundsen, M'Clure and Lancaster), particulary during spring and summer.

Answer: We will revise the abstract and add the comparison in the section of temporal variation of sea ice thickness.

12. L 59-60: This connects back to a previous comment, but when expanding the discussion of shipping through the NWP note the difference between the open water shipping season and potential for ice-breakers. The difference in vessels is critical for shipping along the NWP.

Answer: We appreciate the reviewer's helpful comments. The same answer with Major Comment 1.

13. L 63: revise "we utilize a combination of remotely sensed sea ice concentration data and modelled ice thickness data to examine the sea ice conditions ...."

Answer: Revised.

14. L 68-69: There's a comment below about this selection of a ship path, but I think this needs to be explained in more detail and presented as the "optimal" or "route through the lightest ice conditions".

Answer: We will revise the manuscript.

15. L 74-76: Please elaborate on the description of this dataset. Howe is it collected and what are its limitations. In particular passive microwave data is known to underestimate sea ice concentration during the melt period. This error should be consistent through the time series so it wont dramatically affect your results, but it should be discussed. Regarding the interpretation of sea ice concentration data, you commonly refer to extent, but I believe you are calculating sea ice area. This is good, but the figure labels and text need to be revised. Also please add a sentence about this in the methods. Regarding sea ice concentration data in Prince of Wales Strait – this is a fairly narrow channel and with 25 km resolution I'm wondering how many actual sea ice pixels are contained in this channel and how reliable that data is. Also I haven't seen this channel discussed as part of the NWP before, typically there is one of the

two southern routes or the northern route through M'Clure.

Answer: We will revise the manuscript. We are calculating sea ice extent. The ice extent was calculated by the ice concentration and the control area of each grid. The ice extent was the sum of areas with sea ice concentration greater than 0.15 in the NWP. We will add the formula in the manuscript. The route through Prince of Wales Strait was a modification of north route through M'Clure Strait (Byers and Michael, 2009). Considering it is narrow, few studies have studied it as a separate area, but it has been mentioned in the introduction of the NWP (Sou and Flato, 2009; Pizzolato et al., 2016). In Prince of Wales Strait, there are 16 and 244 sea ice pixels contained from NSIDC and AO-FVCOM, respectively. Reference: Byers, M., & Lalonde, S. (2009). Who controls the northwest passage? Vanderbilt Journal of Transnational Law, 42. Sou, T., & Flato, G. (2009). Sea ice in the Canadian Arctic Archipelago: modeling the past (1950-2004) and the future (2041-60). Journal of Climate, 22(8), 2181-2198. Pizzolato, L., Howell, S. E. L., Dawson, J., Laliberté, Frédéric, & Copland, L. (2016). The influence of declining sea ice on shipping activity in the Canadian Arctic. Geophysical Research Letters, 43(23), 12,146-12,154.

16. L 88: resolution is up to 1 km, but what is the range?

Answer: In the NWP, the range of resolution is from 1 km to 10 km.

17. L 95: It is just ICESat, not ICESat-2.

Answer: Revised.

18. L 94 – 100: In this text please note that all of this work was presented by Zhang et al., (2016). Adjust the start of this text from "We conducted..." to "Zhang et al., (2016) validated ice thickness from AO-FVCOM with a multisource dataset...".

Answer: Revised.

19. L 102-104: Which ERA reanalysis did you use? -Interim or -5? Please specify.
Answer: We used ERA-5 reanalysis data, and we will revise the manuscript.

20. L 108 and throughout: revise the text "significant spatial distribution differences" to " significant differences in the spatial distribution" or "significant spatial differences". As it's written it is tricky to read.

Answer: Revised.

21. L116 – 120: I'm not sure these means are really worth presenting given the significant negative trends you are about to present in the next section. I think the 1979-2017 mean shows the general pattern of ice loss, but I wouldn't get too focused on the actual values.

Answer: We will revise the manuscript.

22. L 120: "After September, the sea ice started to freeze", this is pretty informal, I'd suggest adding some more detail here.

Answer: We will revise the manuscript.

23. L 124-125: Again check that these trends are significant. The Prince of Wales Strait is not significant, so this text needs to be revised. Also just a note that the figures show sea ice extent (area) and the trends are presented. Perhaps provide both the trend in extent and then provide the % for further context.

Answer: We will revise the manuscript.

24. L 126-127: back to one of the major comments, but I am really suspicious of this increasing trend during winter. There may be some variability, but that is likely due to the error of passive microwave sea ice concentration retrievals and not real.

Answer: We will revise the manuscript. The increasing trend in the CAA which was consist with Cavalieri and Parkinson (2008). But considering Cavalieri and Parkinson (2008) also used passive microwave sea ice concentration, we will revise our statement as: Figure 3 exhibited significant differences in the spatial distribution of change rate of

sea ice concentration in the NWP during 1979 to 2017 that the significant decreasing trend occurred in summer and fall and slighter change rate in winter and spring. Reference: Cavalieri, D.J. & Parkinson, C.L. (2008). Antarctic sea ice variability and trends, 1979-2006. J. Geophys. Res. Oceans. 6.

25. L 143 – 147: With respect to Figure 4, the key takeaway is the negative trends in each sub-region. I don't think discussing the mean annual sea ice concentration in each region over the 38 year record is that useful, especially given the substantial changes taking place.

Answer: We will remove the discussion of mean annual sea ice concentration.

26. L 153: Replace "The larger sea ice extent" with "A near complete ice cover..."

Answer: Revised.

27. L 154: The change that occurred around 1997 is related to the 1998 minimum discussed by Howell et al., 2010. Please add the reference of 1998 to the introduction and then you can refer back to it here. Howell, Tivy, Agnew, Markus, Derksen, 2020,Extreme low sea ice years in the Canadian Arctic Archipelago: 1998 versus 2007, GRL.

Answer: We will revise the manuscript.

28. L 163: "Larger sea ice extent resumed in October..." this is a part of fall freeze-up. Beyond listing the regions, I think it would be more useful to state that freeze-up begins earlier in the central and northern part of the CAA (M'Clintock, Peel, Prince of Wales, M'Clure, Viscount, Barrow) in October and then expands to the southern and peripheral part of the CAA during November.

Answer: We will revise the manuscript.

29. L 178: Remove the 2 from ICESat-2.

Answer: Removed.

30. L 182: revise "significant spatial distribution differences"

Answer: Revised.

31. L 183 – 184: As opposed to saying "the sea ice thickness was larger in spring and small in late summer and early fall", simply say "the sea ice was thicker in spring and thinner in later-summer and early fall".

Answer: Revised.

32. L 186 – 187: Thicker ice is located in the Queen Maud Gulf, M'Clintock Channel, and Peel Sound because it operates as a drain trap for Multiyear ice within the CAA as described by Howell et al., 2008. Please add discussion of this to your introduction so you can refer back to it here. Howell, Tivy, Yackel, McCourt (2008), Multi-year sea ice conditions in the western Canadian Arctic Archipelago Region of the northwest passage: 1968-2006, Arctic.

Answer: Revised.

33. L 203 and Figure 8: Again please only display and discuss the significant trends.

Answer: Revised.

34. L 205 – 206: With respect to the increasing trend in the Labrador Strait, first is this significant? If so what is the mechanism for this?

Answer: It is significant in spring and summer. Lancaster Sound was a main pathway for sea ice transport to flow out. During 1978 to 2017, the sea ice area flux showed increasing trend by $7.5 \times 10^3$ km2/10a (Bi et al., 2019) which may result in the thicker MYI from Arctic transporting to Lancaster Sound. We will add the discussion into the manuscript. Reference: Bi, Haibo & Zhang, Zehua & Wang, Yunhe & Xu, Xiuli & Liang, yu & Huang, Jue & Liu, Yilin & Fu, Min. (2019). Baffin Bay sea ice inflow and outflow: 1978–1979 to 2016–2017. The Cryosphere. 13. 1025-1042. 10.5194/tc-13-1025-2019.

35. L 223: "deepest" I think this is in reference back to Figure 1, but the bathymetry of the CAA isn't actually shown in that figure. Please revise or add a reference to this point.

Answer: We will revise the manuscript and add the reference.

36. L 226-228: The process of selecting pathways based on "light sea ice conditions" needs to be revised and considerably improved. Thinner ice doesn't necessarily make a route passable of the ice is still relatively thick and therefore hazardous for all but a few ships. This goes back to the first major comment, a better introduction of shipping along the NWP and the focus on the open water shipping season is needed before this discussion is suitable. Also, if basing the route on the change relative to the historical mean, its important to note the historical mean represents a concentrated, thick ice cover that didn't break up during summer.

Answer: Same answer with Major Comments #1.

37. L263-264: "the low temperatures did not affect the sea ice melting", I'm not sure what is meant by this statement.

Answer: Although SAT showed increasing trend, the temperatures was too low to affect sea ice melting. We will revise it.

38. L 264-265: How are the SST's observed during winter? Are they simply set to be at the freezing point when under sea ice? This should be discussed in the methods and will likely cause this text to be revised.

Answer: We will revise the manuscript.

39. L 311: "Suggest" instead of "suggested"

Answer: Revised.

40. L 311: I'm not sure "residuals" is the right word for what you're describing here. This is the state of the ice cover after the September minimum and at the start of

fall-freeze-up.

Answer: We will revise the statement. It is the averaged value of sea ice concentration and thickness in fall including September, October and November.

41. L 321-323: Connecting the state of the ice cover in fall to the state of the ice cover at the end of winter is interesting, but the last sentence about fall SAT and SST affecting sea ice thickness the following winter is a little misleading, because the state of the ice cover in fall is predominantly dictated by the summer conditions and not just fall.

Answer: We will revise the manuscript.

42. L 325: remove the "ed" from "exerted".

Answer: Revised.

43. L 329: Based on a previous comment, you don't really use observed sea ice thickness data. It's all from the model.

Answer: Revised.

44. L 336 and in other places within the paper: Please be consistent and present sea ice concentration as a percentage (%) as opposed to just "1".

Answer: We appreciate the reviewer's helpful comments. We apologize some statements cause the misunderstanding for the reviewer. We present sea ice concentration as "1". Percentage in trend showed the change rate of sea ice concentration just same as the trend of SAT and SST. We will revise the change rate of sea ice concentration from "%/10a" to "/10a".

45. L 339: Another instance the "increasing trend" during winter. Please check that this is significant.

Answer: Revised.

46. L 343: revise the first sentence to read "from 1979-2017, sea ice thickness in the

NWP decreased..."

Answer: Revised.

47. L 344: Revise "The multiyear mean seasonal sea ice thickness" to read "The monthly mean sea ice thickness..."

Answer: Revised.

48. L 347-348: Remove "the" from "In the most..." and "Lancaster Sound, the sea ice...".

Answer: Revised.

49. L 348: In the conclusions provide a value for these trends.

Answer: We will revise the manuscript.

50. L 361: Revise the word "dominant", the end of winter ice cover is influenced by the fall ice cover, but I don't think it is the dominant factor.

Answer: Removed.

51. Figure 1: Note that you don't actually show bathymetry in the CAA so I would suggest removing bathymetry from this figure.

Answer: Removed (Figure 7).

52. Figure 3: Is the top panel the trends in the annual mean sea ice concentration? I would suggest removing that and focusing on the seasonal means.

Answer: Removed (Figure 1).

53. Figure 4: For the inset of annual cycles can you provide some bounds of the standard deviation or

Answer: Revised (Figure8).

54. Figure 5: This figure can very likely be removed based on comments above.

Answer: Revised.

55. Figure 6: I'd suggest removing "distribution" from the caption as an ice thickness distribution is something other than this figure.

Answer: Revised.

56. Figure 7: Note in the caption that this is the "annual mean thickness".

Answer: Revised.

57. Figure 8: Again, only show the significant trends and consider removing the annual mean.

Answer: Revised (Figure 2).

58. Figure 10: Are these the annual mean changes in SAT and SST? Also, instead of the % please consider revising to show the magnitude ( åŮę C) of the trends and only showing the significant trends.

Answer: These are the annual mean changes in SAT and SST, we revised the title and change unit to " åŮę C/10a" (Figure 3).

59. Figure 11 and 12: It is slightly counter intuitive to flip the colorbar so that blue is positive correlations and red is negative. Also, once again consider showing only the significant correlations.

Answer: Revised (Figure 4, Figure 5).

We are working on the revised manuscript and we will attach a draft revision and highlight the revised places. After getting the comments from other reviewers, we will make further revision. Hopefully, these answers and revisions could meet the reviewer's requirement.

[Figure]

0.03         0.00         −0.03         −0.06         −0.09         −0.12

Change rate (/10a)

Concentration                    Winter

Concentration                    Spring

Concentration                    Summer

Concentration                    Fall

**Fig. 1.** Seasonal mean change rate distribution of sea ice concentration from 1979–2017.

Change rate (m/10a)

Thickness                Winter

Thickness                Spring

Thickness                Summer

Thickness                Fall

**Fig. 2.** Seasonal mean change rate distribution of sea ice thickness from 1979–2017.

[Figure]

**Fig. 3.** Distribution of change rates of SAT and SST in the NWP from 1979 to 2017.

[Figure]

**Fig. 4.** Distribution of seasonal correlation coefficients between sea ice concentration and SAT and SST in the NWP.

[Figure]

**Fig. 5.** Distribution of seasonal correlation coefficients between sea ice thickness and SAT and SST in the NWP.

[Figure]

**Fig. 6.** Comparison of AO-FVCOM sea ice thickness with sea ice thickness observations over the period 1979–2017.

**Fig. 7.** Bathymetry of the Canadian Arctic Archipelago and 10 subregions of Northwest Passage routes.

[Figure]

**Fig. 8.** Annual mean sea ice extent (blue dots) for the NWP and 10 subregions of the NWP from 1979 to 2017. Red lines indicate the linear regression trends.

**Table 1. Mean sea ice thickness, trend and maximum sea ice thickness trend of observations and AO-FVCOM in the Cambridge Bay and Resolute, and mean absolute differences, correlation coefficient, RMSE between observations and AO-FVCOM.**

| | Cambridge Bay | | Resolute | | NWP | |
| --- | --- | --- | --- | --- | --- | --- |
| | Observations | Simulation | Observations | Simulation | CS2SMOS | Simulation |
| Mean sea ice thickness (m) | 1.32 | 1.38 | 1.41 | 1.73 | 1.13 | 1.19 |
| Trend of sea ice thickness (m/10a) | -0.07 (p<0.05) | -0.12 (p<0.01) | 0.03 | 0.00 | -0.02 | -0.09 |
| Trend of maximum sea ice thickness (m/10a) | -0.07 (p<0.05) | -0.12 (p<0.01) | -0.07 (p<0.05) | -0.06 | -0.15 | -0.15 |
| Mean absolute differences (m) | 0.10 | | 0.33 | | 0.20 | |
| Correlation coefficient | 0.96 (p<0.01) | | 0.77 (p<0.01) | | 0.96 (p<0.01) | |
| RMSE (m) | 0.18 | | 0.52 | | 0.25 | |

**Fig. 9.**

---

## Author Comment (AC3) · 10 Oct 2020

Major comments

1. Long-term variation of sea ice and its response to thermodynamic factors in the Northwest Passage of the Canadian Arctic Archipelago author by Shen et al. The manuscript (MS) is interesting and fits with the scope of the journal but unfortunately, the data and the interpretation are not well presented. As the authors have highlighted in the MS title "response to thermodynamic factors", but fail to justify the factors. The authors have discussed only the relation with SST and SAT. To understand thermodynamics, we should know the mixed layer depth (MLD), then only we could know the

ocean heat transport. In my opinion, the article cannot be published in that form needs a lot of substantial improvements and modifications: I, therefore, suggest the article cannot be accepted in the present form.

Answer: We apologize our title of "thermodynamic factors" cause the misunderstanding for the reviewer. The reviewer suggested us to add other two thermodynamic factors including MLD and ocean heat transport into the manuscript. Based on the original title, the study should cover more thermodynamic factors including heat flux and MLD. Since the complicated interactions between sea ice and different thermodynamic factors, the study of each factor needs a lot of analysis, experiments and data support before getting some valuable conclusions. The research of each factor could be discussed in detail as a separated paper. This study mainly focused on the detailed relation of surface atmospheric temperature (SAT), sea surface temperature (SST) with sea ice concentration and thickness in the NWP which examined the spatial difference of impact of SAT and SST on sea ice condition. To make the content clear, we will modify the title to "Long-term variation of sea ice and its response to surface atmospheric temperature and sea surface temperature in the Northwest Passage of the Canadian Arctic Archipelago". Considering the length of article, we will do the study of other thermodynamic factors including MLD and ocean heat transport in the future study.

2. The abstract is very simple and doesn't show the novelty of the pertaining long-term sea ice and its response to thermodynamic factors. After reading the abstract I could see authors have just given the decadal observation of SIC and their correlation with SST and SAT. This section is drafted very poorly with the unfocused aim and finding highlights. Although the approached techniques are good but not justified by the authors in their explanations. Suggested to be more focused and rewrite the abstract. This section is lacking with clear aim and objective of the work as well as the concluding remarks/novelty of the work. Need to be more specific about the computational and processing techniques. The sea ice thickness data for the Canadian Arctic Archipelago were utilized from for the model output of the AO- FVCOM. I could not see any data

validation and any specific reasons for choosing this model. If any such study may be given. Data and methods are not complete need to be elaborated properly. I could not find any analysis details. Sea ice extent data details and analysis are missing how SIE was calculated? How the authors have divided the NWP into 10 subregions? What was the criteria or reference have been considered to divide the CAA? Materials and methods are poorly written and incomplete.

Answer: We appreciate the reviewer's suggestion and have revised the manuscript. We will revise our abstract with more focused aim and finding highlights. For the comment about computational and processing techniques, we have added more detailed descriptions of the analysis methods. The ice extent was calculated by the ice concentration and the control area of each grid. The ice extent was the sum of areas with sea ice concentration greater than 0.15 in the NWP. We will add the formula in the manuscript. For the comment about the validation of model result, we have added the new comparison with observed sea ice thickness of Canadian Ice Services (CIS) and provided exact measure of mean absolute differences, correlation coefficient and RMSE (Table 1, Figure 1). Both the comparison with CIS and CS2SMOS all showed the AO-FVCOM could reasonably reproduce the sea ice thickness in the NWP and the error was less than other models. In addition, the reason we divided the NWP into 10 subregions is based on the map of CAA (Sou and Flato, 2009) and the route of NWP. The NWP was divided into ten subdomains. Reference: Sou, T., & Flato, G. (2009). Sea ice in the Canadian Arctic Archipelago: modeling the past (1950-2004) and the future (2041-60). Journal of Climate, 22(8), 2181-2198.

3. Authors have represented their results in just quantitative way in terms of spatial and temporal changes of SIC, SIE and SIT although sea-ice parameters have been published earlier by several authors sector-wise of whole Arctic regions. The MS is lacking with process and mechanism. Authors have attempted to explain the variations with only SST and SAT, this study needs to be extended by considering ocean heat transport and budget. The sea ice declining processes and their forcings are must be

highlighted. In this present form, the paper is not recommended for publication.

Answer: Similar answer with major comment 1. Through modifying the title to make the present manuscript focus on SAT and SST, we will do further study of other thermodynamic factors in the future work. Hopefully the reviewer could reconsider our manuscript based on the work we have done.

———————————————

[Figure]

[Figure]

**Fig. 1.** Comparison of AO-FVCOM sea ice thickness with sea ice thickness observations over the period 1979–2017.

**Table 1. Mean sea ice thickness, trend and maximum sea ice thickness trend of observations and AO-FVCOM in the Cambridge Bay and Resolute, and mean absolute differences, correlation coefficient, RMSE between observations and AO-FVCOM.**

| | Cambridge Bay | | Resolute | | NWP | |
| --- | --- | --- | --- | --- | --- | --- |
| | Observations | Simulation | Observations | Simulation | CS2SMOS | Simulation |
| Mean sea ice thickness (m) | 1.32 | 1.38 | 1.41 | 1.73 | 1.13 | 1.19 |
| Trend of sea ice thickness (m/10a) | -0.07 ($p<0.05$) | -0.12 ($p<0.01$) | 0.03 | 0.00 | -0.02 | -0.09 |
| Trend of maximum sea ice thickness (m/10a) | -0.07 ($p<0.05$) | -0.12 ($p<0.01$) | -0.07 ($p<0.05$) | -0.06 | -0.15 | -0.15 |
| Mean absolute differences (m) | 0.10 | | 0.33 | | 0.20 | |
| Correlation coefficient | 0.96 ($p<0.01$) | | 0.77 ($p<0.01$) | | 0.96 ($p<0.01$) | |
| RMSE (m) | 0.18 | | 0.52 | | 0.25 | |

**Fig. 2.**